# Homophily-Heterogeneity Gradient Surgery for Federated Graph Learning

**Sujia Huang**[1]  **Lele Fu**[2]  **Shunxin Xiao**[3]  **Xiaoya Zhang**[1]  **Chunyan Xu**[1]  **Tong Zhang**[1]  **Bo Huang**[1]  **Zhen Cui**[4]

## Abstract

Federated Graph Learning (FGL) facilitates privacy-preserving collaborative training of graph neural networks, yet homophily heterogeneity across subgraphs can induce optimization conflicts that degrade model generalization. Many existing solutions rely on multi-channel architectures to mitigate such conflicts, which increase the burden on edge devices and lack theoretical convergence analysis. To overcome these limitations, we propose FedGCM, a novel FGL framework with Group-oriented Conflict Mitigation, which aligns inconsistent optimization objectives via a tailored gradient surgery. Specifically, FedGCM first divides clients into distinct groups based on their homophily levels, thereby avoiding exhaustive client-to-client conflict assessments. To resolve inter-group interference, we develop RP-Grad, a gradient surgery mechanism based on residual projection, which integrates synergistic knowledge while filtering inter-group conflicts. The refined updates are then transmitted in a group-wise fashion, effectively alleviating optimization conflicts induced by homophily heterogeneity without augmenting the client-side burden. Furthermore, we provide a formal theoretical analysis establishing the convergence. Extensive experiments on both homophilous and heterophilous graphs demonstrate that FedGCM consistently achieves superior performance.

[1]School of Computer Science and Engineering, Nanjing University of Science and Technology, Nanjing, China [2]School of Systems Science and Engineering, Sun Yat-Sen University, Guangzhou, China [3]School of Computer and Information Engineering, Fujian Key Laboratory of Pattern Recognition and Image Understanding, Xiamen University of Technology, Xiamen, China [4]School of Artificial Intelligence, Beijing Normal University, Beijing, China. Correspondence to: Zhen Cui <zhen.cui@bnu.edu.cn>, Tong Zhang <tong.zhang@njust.edu.cn>, Lele Fu <lawrence-fzu@gmail.com>.

*Proceedings of the 43rd International Conference on Machine Learning*, Seoul, South Korea. PMLR 306, 2026. Copyright 2026 by the author(s).

## 1. Introduction

Graphs serve as a universal abstraction for modeling complex relational dependencies, empowering a diverse spectrum of domains, including biomedicine (Bang et al., 2023; Zhang et al., 2025), social network (Wang et al., 2025a; Zhang, 2025), intelligent transportation systems (Wang et al., 2022; Chen et al., 2024), and others (Yu et al., 2023; Zhuo et al., 2024b; Kou et al., 2024; Zhu et al., 2026). For processing graph-structured data, Graph Neural Networks (GNNs) have emerged as a dominant paradigm, achieving state-of-the-art performance by effectively integrating topological information with node features (Luan et al., 2022; Chen et al., 2025b; Zhuo et al., 2023; 2024a; Huang et al., 2026). However, the training of high-performance GNNs requires large-scale datasets, which, in practice, are frequently fragmented across disparate edge devices or institutions. Furthermore, stringent privacy protocols often prohibit the direct consolidation of raw data, resulting in data silos. To bridge this gap, Federated Learning (FL) has gained prominence as a privacy-preserving collaborative paradigm (Hanser et al., 2025; Li et al., 2025b; Kou et al., 2026; Qi et al., 2026), enabling distributed model training through the exchange of model-related information rather than raw data. Within this context, Federated Graph Learning (FGL) has been developed to orchestrate multi-client collaboration, specifically tailored to learn global GNN models from sensitive graph data (Baek et al., 2023; Fu et al., 2024; Wan et al., 2025).

Nevertheless, conventional GNNs are typically designed under the homophily assumption, which posits that connected nodes tend to share the same labels. This assumption can lead to performance degradation in heterophilous scenarios, where connected nodes frequently belong to distinct categories. Although numerous heterophilous GNNs (Li et al., 2025a; Huang et al., 2025; Zhao et al., 2025) have been proposed for single-graph environments, their effectiveness is challenged in FGL due to homophily heterogeneity across clients, i.e., substantial variations in homophily levels across local subgraphs (Tan et al., 2025). Such heterogeneity induces divergent local optimization preferences: homophilous clients benefit from smoothing features over local neighborhoods, whereas heterophilous clients require the model to retain discriminative signals among semantically dissimilar neighbors. Neglecting these inconsistencies

can trigger optimization conflicts during global aggregation, thereby degrading the performance of the global model. To address this, recent algorithms have explored multi-channel learning paradigms, which independently maintain and collaboratively optimize representation spaces designed to distinct topological patterns (Yu, 2025; Chen et al., 2025a; Tan et al., 2025), as shown in Figure 1(a).

Despite these advancements, many FGL participants operate under resource constraints; maintaining concurrent model branches *substantially increases the parameter count and exacerbates memory overhead*. Meanwhile, the requirement to synchronize multiple sets of parameters escalates communication costs, potentially causing congestion and latency in bandwidth-limited environments. Furthermore, most of them *lack theoretical convergence analysis*, making it challenging to formally characterize their optimization behavior during collaborative training on distributed graph data.

To address the aforementioned limitations, we propose FedGCM, a theoretically grounded Federated graph learning framework with Group-oriented Conflict Mitigation. Unlike prior multi-branch architectures, FedGCM mitigates optimization conflicts through inter-group gradient surgery while maintaining a streamlined architecture on the client side, as shown in Figure 1(b). Specifically, to overcome label inaccessibility in semi-supervised settings, we move beyond conventional label-centric metrics. Instead, we estimate each client's homophily level by exploiting the positive correlation between the high-frequency area of graph signals and graph heterophily. This enables a homophily-driven grouping strategy that clusters clients into distinct cohorts along the homophily–heterophily spectrum, eliminating the need for exhaustive pairwise conflict detection. Subsequently, to resolve inter-group interference, we design RPGrad, a residual-projection-based gradient surgery mechanism. When gradients from auxiliary groups exhibit directional similarity, RPGrad incorporates their synergistic components to enhance knowledge sharing; otherwise, it performs orthogonal projection to suppress conflicting components caused by directional discrepancies. In addition, the theoretical demonstration provides convergence analysis for RPGrad, offering formal guarantees for its conflict-aware collaborative optimization. Notably, during communication, the server broadcasts group-oriented updates according to predefined group indices, thereby avoiding additional computational or communication overhead on resource-constrained clients.

Our primary contributions are summarized as follows:

- **We address homophily-heterogeneity-induced optimization conflicts.** We propose FedGCM, a gradient-surgery-based framework that mitigates such conflicts without incurring the architectural burdens typical of multi-channel approaches.

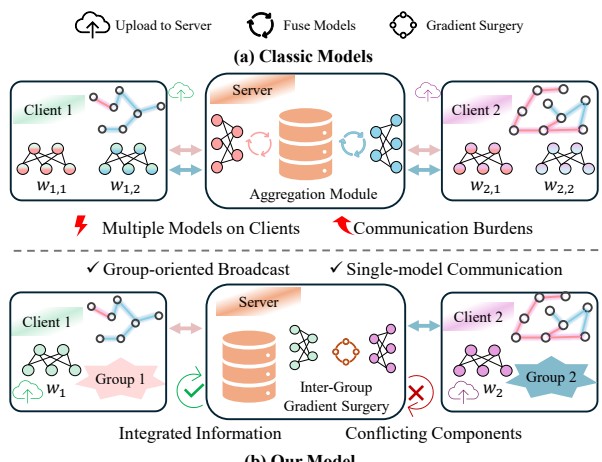

*Figure 1.* Comparison of FGL paradigms for addressing homophily heterogeneity. (a) Prior multi-channel frameworks, which increase client-side memory and communication overhead; (b) Our FedGCM, which retains a standard single-model client architecture through server-side inter-group gradient optimization.

- **We develop a gradient surgery mechanism with the residual projection.** The proposed RPGrad performs inter-group optimization by excising conflicting directions and integrating knowledge across groups, with theoretical convergence analysis.

- Comprehensive experimental results on diverse homophilous and heterophilous benchmarks demonstrate that FedGCM achieves state-of-the-art performance, significantly outperforming competitive baselines.

## 2. Related Work

### 2.1. Federated Learning

FL enables collaborative model training across clients without direct data sharing (Wang et al., 2025b; Lin et al., 2025). However, the Non-Independent and Identically Distributed (Non-IID) nature of local data poses a fundamental challenge, manifesting primarily as diverse feature and label distributions. To mitigate these discrepancies, FedNova (Wang et al., 2020) focuses on objective inconsistency, MOON (Li et al., 2021) uses model-level contrastive learning, and Fed-Prox (Li et al., 2020a) introduces a proximal term to ensure convergence. To further capture cross-client variances, Personalized Federated Learning (PFL) has garnered significant attention, which focuses on balancing global knowledge aggregation with the preservation of local expertise (Yang et al., 2023; Zhu et al., 2025). Representative approaches like FedPer (Arivazhagan et al., 2019) and FedALA (Zhang et al., 2023) customize local models via layer decoupling or adaptive weight aggregation, respectively. Building upon the success of FL, federated graph learning has emerged as a promising paradigm for processing distributed graph data.

## 2.2. Federated Graph Learning under Homophily Heterogeneity

FGL focuses on collaboratively training GNNs to capture complex graph topologies (Fu et al., 2025; Long et al., 2025; Wu et al., 2026). For example, FedPUB (Baek et al., 2023) utilizes random graphs to compute client similarities. Fed-TAD (Zhu et al., 2024) decouples node and topology variation to capture subgraph heterogeneity. FedGTA (Li et al., 2023) designs a topology-aware optimization strategy. Stemming from diverse homophily levels across clients, FGL faces a challenge named homophily heterogeneity. Such heterogeneity causes local models to drift toward conflicting optimization directions, resulting in suboptimal global performance (Li et al., 2024; Yu, 2025). To mitigate this issue, prevailing methods employ multi-channel architectures to decouple message passing. For instance, FedSPA (Tan et al., 2025) decomposes the global graph into homophilous, heterophilous, and unknown subgraphs, respectively, while FedHERO (Chen et al., 2025a) adopts a dual-channel framework to separate latent global structures from personalized information. Despite their promise, these approaches often impose substantial computational and communication burdens on edge devices due to their complex architectural designs. To overcome these limitations, we propose FedGCM, which mitigates optimization conflicts without increasing client-side resource costs.

## 3. Methodology

### 3.1. Preliminary

#### 3.1.1. NOTATIONS

Let $\mathcal{G} = (\mathcal{V}, \mathcal{E})$ be a graph with $|\mathcal{V}| = N$ nodes and $|\mathcal{E}| = E$ edges. Node features are denoted by a matrix $\mathbf{X} \in \mathbb{R}^{N \times D}$, where the $v$-th row $\mathbf{x}_v \in \mathbb{R}^D$ indicates the $D$-dimensional attribute vector of node $v$. The adjacency matrix is $\mathbf{A} \in \{0,1\}^{N \times N}$, where $A_{uv} = 1$ if $(u,v) \in \mathcal{E}$, and 0 otherwise. Let $\hat{\mathbf{A}} = \mathbf{D}^{-1/2}(\mathbf{A} + \mathbf{I})\mathbf{D}^{-1/2}$ denote the symmetric normalized adjacency matrix with the degree matrix $\mathbf{D}$ of $(\mathbf{A} + \mathbf{I})$ and the identity matrix $\mathbf{I}$. The symmetric normalized Laplacian matrix is subsequently defined as $\mathbf{L} = \mathbf{I} - \hat{\mathbf{A}}$. The ground-truth label matrix is denoted by $\mathbf{Y} \in \{0,1\}^{N \times C}$, where each row $\mathbf{y}_v \in \mathbb{R}^C$ is a one-hot vector across $C$ classes. The node set $\mathcal{V}$ is partitioned into a labeled subset $\mathcal{V}_{lab}$ and an unlabeled subset $\mathcal{V}_{unlab}$, where the objective is to predict labels for $\mathcal{V}_{unlab}$ by training a model supervised by the nodes in $\mathcal{V}_{lab}$.

#### 3.1.2. FL-ASSISTED GNNS

As the foundational strategy, FedAvg (McMahan et al., 2017) serves as the backbone of various FL algorithms. We delineate the standard FGL training procedure utilizing FedAvg as follows. Consider a centralized server orches-

trating $M$ clients, where the global graph is distributed into $M$ local subgraphs $\{\mathcal{G}_i = (\mathbf{A}_i, \mathbf{X}_i)\}_{i=1}^M$. Each client $i$ performs downstream tasks on its local topology $\mathbf{A}_i$ and feature matrix $\mathbf{X}_i$, followed by server-side aggregation and redistribution of model parameters. Specifically, at each communication round $t$, the $i$-th client receives the global model $w^t$ from the server and conducts local optimization on its private dataset, i.e.,

$$w_i^t = w^t - \tau^t g_i^t, \tag{1}$$

where $w^t$ and $w_i^t$ denote the global and local models, respectively, $\tau^t$ is the learning rate, and $g_i^t$ is the local gradient. In the semi-supervised node classification, the local objective $\mathcal{L}_i$ of client $i$ is instantiated by the Cross-Entropy loss. Given a GNN-based classifier $f(\cdot)$, the loss over the local labeled node set $\mathcal{V}_{i,lab}$ is formulated as:

$$\mathcal{L}_i(w) = - \sum_{u \in \mathcal{V}_{i,lab}} \sum_{c=1}^C Y_{uc} \ln f_{uc}(\mathbf{A}_i, \mathbf{X}_i; w). \tag{2}$$

For the server, it receives the updated weights from all participating clients and computes the new global model by:

$$w^{t+1} = \sum_{i=1}^M \frac{N_i}{N} w_i^t, \tag{3}$$

where $N_i$ is the number of samples at client $i$.

#### 3.1.3. GRAPH-LEVEL HOMOPHILY

For a graph $\mathcal{G}$ with label matrix $\mathbf{Y}$, graph homophily $H$ quantifies the label consistency among connected nodes. This is typically measured via node-level ($H_n$) or edge-level ($H_e$) metrics:

$$\begin{aligned} H_n &= \frac{1}{N} \sum_{v \in \mathcal{V}} \frac{|\{u \in \mathcal{N}_v | \mathbf{y}_u = \mathbf{y}_v\}|}{|\mathcal{N}_v|}, \\ H_e &= \frac{|\{(u,v) \in \mathcal{E} | \mathbf{y}_u = \mathbf{y}_v\}|}{|\mathcal{E}|}, \end{aligned} \tag{4}$$

where $\mathcal{N}_v$ is the neighborhood of node $v$. In practice, however, label scarcity in semi-supervised settings restricts the direct computation of $H$. Following (Tang et al., 2022; Tan et al., 2025), we use the spectral properties of graph signals as an unsupervised proxy for graph homophily. Specifically, let $\mathbf{L}$ be the Laplacian matrix with the eigen-decomposition $\mathbf{L} = \mathbf{U}\mathbf{\Gamma}\mathbf{U}^\top$, where $\mathbf{\Gamma} = \text{diag}(\lambda_1, \cdots, \lambda_N)$ contains the eigenvalues $0 = \lambda_1 \leq \cdots \leq \lambda_N \leq 2$ and the matrix $\mathbf{U}$ consisted of eigenvectors. The Fourier transform of graph signal $\mathbf{x}$ is given by $\hat{\mathbf{x}} = \mathbf{U}^\top \mathbf{x}$. Then, we have

$$S = \frac{\sum_{k=1}^N \lambda_k \hat{\mathbf{x}}^2}{\sum_{k=1}^N \hat{\mathbf{x}}^2} = \frac{\mathbf{x}^\top \mathbf{L} \mathbf{x}}{\mathbf{x}^\top \mathbf{x}} \propto (1 - H). \tag{5}$$

$S$ is the high-frequency area of graph signal, which captures signal variations over the graph topology and is inversely correlated with homophily $H$ (Tan et al., 2025). This facilitates label-free homophily profiling using raw features.

### 3.2. The Proposed Model

In this subsection, we detail the proposed FedGCM, whose overall architecture is illustrated in Figure 2. The server first partitions clients into distinct groups, then performs gradient surgery to integrate synergistic knowledge while avoiding conflicts, and finally broadcasts group-oriented updates to clients. FedGCM includes two crucial components: 1) Homophily-driven grouping, which partitions clients into $K$ distinct cohorts based on their high-frequency areas inversely correlated with local homophily levels; 2) Inter-group gradient surgery (RPGrad), a residual-projection-based optimization mechanism that removes conflicting gradient components while preserving synergistic information across groups.

#### 3.2.1. HOMOPHILY-DRIVEN GROUPING

Given the limited availability of labels, the overall homophily level for each client cannot be directly estimated through conventional label-dependent metrics. To circumvent this, we alternatively utilize the high-frequency area of the graph spectrum. As illustrated in Eqn. (5), this spectral signature quantifies the fluctuations of graph signal across the topology, which are inversely correlated with the homophily level $H$. For client $i$, the dimension-wise high-frequency area is formulated as:

$$S_{i,d} = \frac{[\mathbf{X}_i]_{:d}^\top \mathbf{L}_i [\mathbf{X}_i]_{:d}}{[\mathbf{X}_i]_{:d}^\top [\mathbf{X}_i]_{:d}}. \qquad (6)$$

Here, $\mathbf{L}_i$ is the local Laplacian derived from the client's subgraph $\mathbf{A}_i$, and $[\mathbf{X}_i]_{:d}$ is the $d$-th column of the feature matrix $\mathbf{X}_i$, representing the feature vector at the $d$-th dimension across all nodes belonging to client $i$. A large $S_{i,d}$ represents that the $d$-th feature varies sharply between neighboring nodes, reflecting weak local smoothness. $S_i$ is therefore obtained by averaging the differences over all $D$ feature dimensions, i.e., $S_i = \frac{1}{D}\sum_{d=1}^{D} S_{i,d}$.

To mitigate the overhead of reconciling divergent optimization preferences among all client pairs, we partition the participating clients into $K$ distinct groups based on their $S_i$. For client $i$, the group assignment index $I(i) \in \{0, \ldots, K-1\}$ is determined by

$$I(i) = \min\left(\left\lfloor \frac{S_i - S_{\min}}{S_{\max} - S_{\min} + \epsilon} \times K \right\rfloor, K-1\right), \quad (7)$$

where $S_{\max}$ and $S_{\min}$ denote the extremum values of the high-frequency area among the current participants, $\lfloor \cdot \rfloor$ represents the floor operator, and $\epsilon$ is a small constant to prevent division by zero. This strategy transforms the global gradient conflicts into inter-group optimization problem, thereby reducing the overhead of conflict detection and mitigation under large-scale client participation. Given the negative correlation between the high-frequency area $S$ and the homophily level $H$, the ascending group index $I(i)$ inherently reflects a topological transition from homophily-dominant to heterophily-dominant structures.

Rather than pursuing a global consensus that may lead to sub-optimal average performance (Lu et al., 2022), FedGCM adopts a group-oriented optimization paradigm to foster group-wise knowledge specialization. Formally, the optimization objective for the $k$-th group can be formulated as:

$$\min_{\widetilde{w}_k} \mathcal{L}_k(\widetilde{w}_k), \quad \mathcal{L}_k(\widetilde{w}_k) = \sum_{i \in \mathcal{K}_k} \frac{N_i}{N_k} \mathcal{L}_i(\widetilde{w}_k), \qquad (8)$$

where $\mathcal{K}_k = \{i | I(i) = k\}$ denotes the index set of clients belonging to the $k$-th group, $N_k = \sum_{i \in \mathcal{K}_k} N_i, N_k > 0$ and $\widetilde{w}_k$ represents the group-specific model parameters.

> *Remark* 3.1. The presence of vacant groups during the partitioning process does not hinder the optimization and inter-group knowledge interaction of remaining groups. In addition, in scenarios where subgraphs exhibit high structural congruence or in small-scale deployments, all invited clients may be assigned to a single group. Under such conditions, we consider that the optimization directions are effectively aligned, and the proposed FedGCM naturally reduces to standard GNN-based FedAvg. This property underscores the framework's flexibility in handling extreme data distributions.

#### 3.2.2. GROUP-ORIENTED GRADIENT SURGERY

Following the partitioning procedures delineated above, we obtain the group assignments $\{I(i)\}_{i=1}^M$ for all clients, which can be viewed as a fixed structural attribute. In each communication round $t$, clients transmit their local models $\{w_i^t\}_{i=1}^M$ to the central server[1]. To effectively mitigate the optimization conflicts stemming from homophily heterogeneity across subgraphs, we propose an inter-group gradient surgery scheme based on residual projection, termed RPGrad. We first formalize the concept of gradient conflict, which is recognized as a direct cause of performance degradation in FL (Wang et al., 2021; Pan et al., 2025).

**Definition 3.2.** Let $g_i$ and $g_j$ denote the local gradients of clients $i$ and $j$, respectively. The gradients are considered

---

[1]Since the server can persist the group indices after the initial synchronization, for any subsequent communication round $t > 0$, clients only upload their model parameters $\{w_i^t\}_{i=1}^M$ without incurring additional communication overhead.

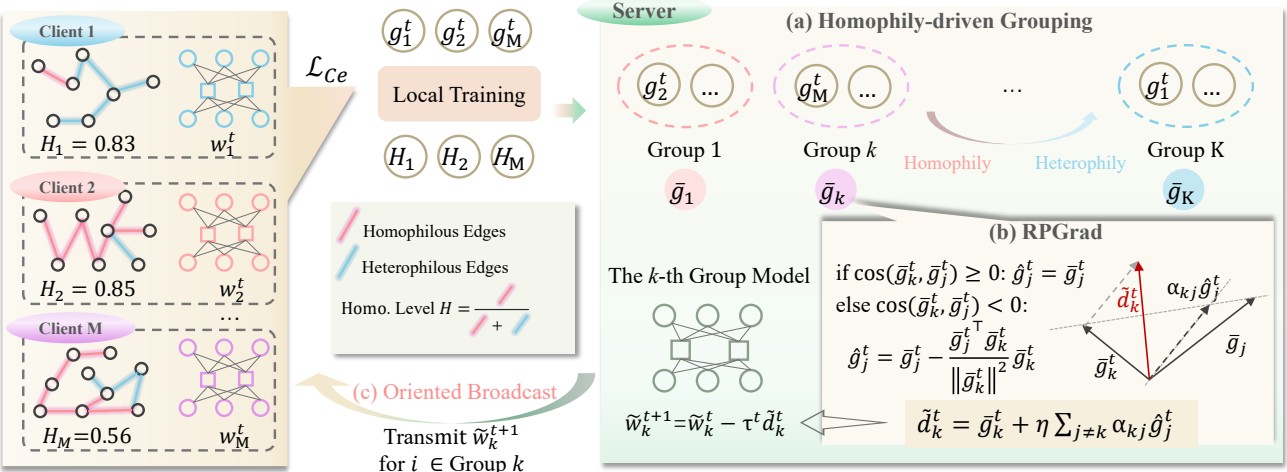

*Figure 2.* Architectural overview of FedGCM. Clients execute local training and parameter synchronization via standard FL protocols. The server clusters clients into $K$ homophily-aware groups and computes refined gradient directions $\{\widetilde{d}_k^t\}_{k=0}^{K-1}$ utilizing inter-group gradient surgery with residual projection. Finally, group-oriented updates are broadcast to the corresponding clients.

to be in conflict with each other when the cosine similarity between them is negative, i.e., $cos(g_i, g_j) < 0$.

To examine whether gradients from different groups may point in divergent directions during training, we compute the cosine similarity between gradients across groups on two homophilous datasets (Cora and CiteSeer) and two heterophilous datasets (Chameleon and Minesweeper). As shown in Table 1, with $K = 4$, several group pairs exhibit negative cosine similarity, indicating the presence of inter-group gradient conflicts. Overall, these statistics reveal that the gradient directions across groups are diverse, providing empirical evidence for the significance of considering inter-group optimization conflicts.

*Table 1.* Cosine similarity between gradients across groups.

| Datasets | G0-1 | G0-2 | G0-3 | G1-2 | G1-3 | G2-3 |
|---|---|---|---|---|---|---|
| Chameleon | -0.045 | 0.043 | 0.226 | -0.065 | -0.120 | 0.156 |
| Minesweeper | 0.274 | -0.144 | 0.035 | 0.157 | -0.144 | -0.205 |
| Cora | -0.013 | -0.111 | -0.167 | 0.067 | 0.066 | 0.124 |
| CiteSeer | 0.150 | 0.086 | 0.059 | 0.006 | -0.005 | 0.126 |

A classic approach for mitigating gradient conflicts is PC-Grad (Yu et al., 2020), which iteratively projects the gradient of a target task onto the normal plane of any conflicting gradient. Inspired by this, we introduce RPGrad, a gradient correction mechanism designed to absorb non-conflicting information from other groups via residual projection while safeguarding the primary optimization direction. Specifically, for each group $k$, we first calculate its average gradient at round $t$ as: $\bar{g}_k^t = \sum_{i \in \mathcal{K}_k} \frac{N_i}{N_k} g_i^t$ with $g_i^t = (\widetilde{w}_k^t - w_i^t)/\tau^t$. The refined gradient direction $\widetilde{d}_k^t$ for the $k$-th group is then

derived by augmenting its group-specific gradient with a residual collaboration term, formulated as a weighted sum of rectified gradients from other groups:

$$\widetilde{d}_k^t = \bar{g}_k^t + \eta \sum_{\substack{j=0 \\ j \neq k}}^{K-1} \alpha_{kj}^t \hat{g}_j^t \quad \begin{cases} \hat{g}_j^t = \bar{g}_j^t, & \text{if } \cos\left(\bar{g}_k^t, \bar{g}_j^t\right) \geq 0, \\ \hat{g}_j^t = \bar{g}_j^t - \frac{\bar{g}_j^{t\top} \bar{g}_k^t}{\|\bar{g}_k^t\|^2} \bar{g}_k^t, & \text{otherwise,} \end{cases}$$

(9)

where $\eta > 0$ is a trade-off hyperparameter controlling the strength of cross-group collaboration. The collaboration weight $\alpha_{kj}^t$ is determined by the directional similarity between groups, followed by Softmax-normalization

$$\alpha_{kj}^t = \frac{\exp\left(\cos(\bar{g}_k^t, \bar{g}_j^t)\right)}{\sum_{m=0, m \neq k}^{K-1} \exp\left(\cos(\bar{g}_k^t, \bar{g}_m^t)\right)}.$$

(10)

RPGrad ensures that $\bar{g}_k^t \cdot \widetilde{d}_k^t > 0$ (detailed proof provided in Appendix A). This property guarantees that the corrected gradient direction $\widetilde{d}_k^t$ maintains a positive alignment with the original group gradient $\bar{g}_k^t$. Following the computation of $\widetilde{d}_k^t$, the server executes group-oriented broadcasting, and updates the parameters for both the server and clients as follows:

$$\begin{aligned} \textbf{Server:} \quad & \widetilde{w}_k^{t+1} = \widetilde{w}_k^t - \tau^t \widetilde{d}_k^t, \\ \textbf{Client } i\textbf{:} \quad & w_i^{t+1} = \widetilde{w}_k^{t+1} - \tau^t g_i^{t+1}, \end{aligned}$$

(11)

where $\widetilde{w}_k^t$ and $w_i^t$ represent the $k$-th group model and the local model of client $i \in \mathcal{K}_k$, respectively. Algorithm 1 summarizes the complete procedure of the proposed framework.

**Algorithm 1** FedGCM

1: **Input:** Global graph $\mathcal{G} = (\mathbf{X}, \mathbf{A})$, initial learning rate $\tau^0$, client size $M$ and group size $K$, communication rounds $T$ and hyperparameter $\eta$.
2: Initialize server parameters $\{\widetilde{w}_k^0\}_{k=0}^{K-1}$ and distribute subgraphs $\{\mathcal{G}_i = (\mathbf{X}_i, \mathbf{A}_i)\}_{i=1}^M$ to respective clients;
3: Compute the high-frequency area $S_i$ of client $i$ using Eqn. (6);
4: Obtain the group index $I(i) \in \{0, \ldots, K-1\}$ of client $i$ with Eqn. (7);
5: **for** $t = 0, \ldots, T-1$ **do**
6:     Server sends group-specific model $\widetilde{w}_{I(i)}^t$ to client $i$;
7:     $w_i^t \leftarrow$ Client $i$ performs local training with Eqn. (2);
8:     Derive local gradient $g_i^t = (\widetilde{w}_{I(i)}^t - w_i^t)/\tau^t$;
9:     **for** each group $k = 0, \ldots, K-1$ **do**
10:         Compute average gradient $\bar{g}_k^t = \sum_{i \in \mathcal{K}_k} \frac{N_i}{N_k} g_i^t$;
11:         Obtain update direction $\widetilde{d}_k^t$ via Eqn. (9);
12:         Update group parameter: $\widetilde{w}_k^{t+1} \leftarrow \widetilde{w}_k^t - \tau^t \widetilde{d}_k^t$;
13:     **end for**
14: **end for**
15: **Return:** Models $\{\widetilde{w}_k\}_{k=0}^{K-1}$;

### 3.2.3. CONVERGENCE ANALYSIS

In this subsection, we provide a convergence analysis of the proposed algorithm under a convex surrogate setting.

We first adhere to the following standard assumptions (Li et al., 2020b; Reddi et al., 2020):

**Assumption 3.3.** The objective function $\mathcal{L}_k$ is convex, satisfying the following inequality for any model state $\widetilde{w}_k^t$ and the optimal solution $\widetilde{w}_k^*$:

$$\mathcal{L}_k(\widetilde{w}_k^t) - \mathcal{L}_k(\widetilde{w}_k^*) \leq \langle \nabla \mathcal{L}_k(\widetilde{w}_k^t), \widetilde{w}_k^t - \widetilde{w}_k^* \rangle. \quad (12)$$

**Assumption 3.4.** The norms of both the gradient $\nabla \mathcal{L}_k(\widetilde{w}_k^t)$ and the update direction $\widetilde{d}_k^t$ are uniformly bounded by a constant $G \geq 0$.

**Lemma 3.5.** *There exists a constant $\delta \geq 0$ such that the deviation between the update direction $\widetilde{d}_k^t$ and the gradient $\nabla \mathcal{L}_k(\widetilde{w}_k^t)$ is bounded by:*

$$\|\widetilde{d}_k^t - \nabla \mathcal{L}_k(\widetilde{w}_k^t)\|_2 \leq \delta. \quad (13)$$

Based on Assumptions 3.3-3.5, we present the following theorem:

**Theorem 3.6.** *Let $\widetilde{w}_k^*$ be an optimal solution for the $k$-th group objective. For any positive sequence of step sizes $\{\tau^t\}_{t=1}^T$, the minimum optimization gap after $T$ rounds is bounded by:*

$$\min_{t=1,\ldots,T} \left( \mathcal{L}_k(\widetilde{w}_k^t) - \mathcal{L}_k(\widetilde{w}_k^*) \right) \leq \frac{E^2 + G^2 \sum_{t=1}^T (\tau^t)^2}{2 \sum_{t=1}^T \tau^t} + \delta R, \quad (14)$$

*where $E = \|\widetilde{w}_k^1 - \widetilde{w}_k^*\|_2$ is the distance between the initial parameters and the optimal solution and $R$ is the radius of the parameter search space, i.e., $\|\widetilde{w}_k^t - \widetilde{w}_k^*\|_2 \leq R$.*

*Proof.* The detailed proof is provided in Appendix A. □

*Remark* 3.7. By adopting a diminishing step-size strategy such as $\tau^t \propto \frac{1}{\sqrt{t}}$, the numerator of the bound in Eqn. (14) grows logarithmically ($\sum (\tau^t)^2 \propto \ln T$), while the denominator grows at a faster rate ($\sum \tau^t \propto \sqrt{T}$). Consequently, the first term vanishes as $T \to \infty$, leading to $\lim_{T \to \infty} (\min_{t=1,\ldots,T} (\mathcal{L}_k(\widetilde{w}_k^t) - \mathcal{L}_k(\widetilde{w}_k^*))) \leq \delta R$, which indicates that the best-iterate optimization gap converges to a bounded $\delta R$-neighborhood of the optimal objective value.

## 4. Experiments

In this section, we conduct a comprehensive empirical evaluation of the proposed FedGCM. Specifically, we compare FedGCM with competitive baselines, analyze its parameter sensitivity, conduct ablation studies, and examine its convergence behavior. Additional experimental results are provided in Appendix D.

### 4.1. Experimental Settings

Datasets. We evaluate the effectiveness of FedGCM across nine diverse graph benchmark datasets. Specifically, the evaluation includes four homophilous graphs: three citation networks (Cora, CiteSeer, PubMed) (Yang et al., 2016) and the Coauthor-Physics network (Shchur et al., 2018). Furthermore, we incorporate five heterophilous datasets: two Wikipedia-based networks (Chameleon and Squirrel) (Pei et al., 2020), the synthetic Minesweeper graph, the product co-purchasing network Amazon-ratings (Platonov et al., 2023), and the Penn94 social network (Lim et al., 2021). Comprehensive statistics and detailed descriptions of these datasets are provided in Appendix C.

Baselines. We evaluate FedGCM against ten state-of-the-art baselines to provide a comprehensive performance comparison. These benchmarks include FedAvg (McMahan et al., 2017), the foundational federated learning algorithm, implemented with two distinct backbones: FedAvg$_{GCN}$ (standard GCN) and FedAvg$_{ACM}$ (ACM-GNN for heterophily). Additionally, we include FedNova (Wang et al., 2020) to address the objective inconsistency problem in traditional FL, and eight FGL frameworks: FedPUB (Baek et al., 2023), FedGTA (Li et al., 2023), AdaFGL (Li et al., 2024), FedTAD (Zhu et al., 2024), FedIIH (Yu et al., 2025), FedHERO (Chen et al., 2025a), FedGSP (Yu, 2025) and FedSPA (Tan et al., 2025), where AdaFGL, FedHERO, FedGSP and FedSPA are designed to homophily heterogeneity.

Implement Details. To simulate a federated environment,

*Table 2.* Node classification performance on homophilous and heterophilous datasets: Mean ACC % (Standard Deviation %). The first- and second-best results are highlighted in **bold** and underlined, respectively. OoM means that methods suffer from out-of-memory error.

| Methods | Homophilous Datasets | | | | Heterophilous Datasets | | | | |
|---|---|---|---|---|---|---|---|---|---|
| | Cora | CiteSeer | PubMed | Physics | Chameleon | Squirrel | Minesweeper | Amazon-ratings | Penn94 |
| $FedAvg_{GCN}$ | 84.7 (0.5) | 71.4 (0.9) | 85.1 (0.2) | 86.7 (0.5) | 38.7 (0.9) | 26.7 (1.1) | 81.2 (0.4) | 47.7 (0.5) | 65.9 (0.5) |
| $FedAvg_{ACM}$ | 84.3 (0.3) | 69.5 (0.3) | 87.6 (0.3) | 93.0 (2.3) | 61.1 (1.1) | 44.5 (1.8) | 81.6 (0.1) | 49.5 (0.5) | 72.9 (0.7) |
| FedNova | 83.4 (0.8) | 70.5 (0.9) | 87.2 (0.1) | 94.2 (0.8) | 62.5 (2.2) | 47.0 (1.9) | 84.2 (0.4) | 49.4 (0.2) | 72.1 (0.4) |
| FedPUB | 83.6 (0.7) | 71.4 (0.9) | 87.1 (0.3) | 95.6 (0.1) | 63.1 (1.0) | 45.8 (1.7) | 84.1 (0.5) | 49.9 (0.1) | 72.1 (1.6) |
| FedGTA | 85.5 (0.4) | 72.5 (0.5) | 87.7 (0.2) | 95.6 (0.1) | 61.3 (1.2) | 45.8 (1.3) | 83.2 (0.1) | 52.2 (0.6) | 73.4 (0.8) |
| AdaFGL | 83.5 (0.0) | 73.0 (0.0) | 87.5 (0.0) | 94.8 (0.7) | 64.3 (0.1) | 48.4 (1.9) | 84.3 (0.1) | 52.5 (0.1) | 75.5 (0.2) |
| FedTAD | 81.8 (0.5) | 72.8 (1.0) | 87.9 (0.3) | 95.1 (1.7) | 64.1 (2.9) | 47.0 (1.9) | 84.8 (0.3) | 53.0 (0.3) | 75.3 (1.1) |
| FedIIH | 82.0 (0.1) | 70.9 (0.6) | 88.0 (0.4) | 93.4 (0.5) | 61.8 (1.6) | 48.7 (0.2) | 80.6 (0.4) | 48.9 (1.6) | 71.1 (0.5) |
| FedHERO | 85.0 (1.3) | 69.4 (1.7) | 87.4 (0.7) | 95.8 (0.2) | 61.6 (1.4) | 46.2 (3.0) | 83.9 (0.4) | 49.1 (0.4) | 73.8 (2.6) |
| FedGSP | 85.7 (0.6) | 70.8 (0.5) | 87.3 (0.4) | 95.1 (0.5) | 62.2 (1.4) | 45.9 (4.1) | 81.5 (0.1) | 47.4 (0.7) | OoM |
| FedSPA | 84.9 (0.7) | 72.5 (0.7) | 87.9 (0.3) | 95.3 (0.3) | 62.3 (1.9) | 47.5 (2.4) | 83.2 (0.2) | 50.3 (0.5) | 74.1 (0.3) |
| Ours | **86.1 (0.6)** | **74.6 (0.4)** | **88.5 (0.2)** | **96.1 (0.1)** | **68.6 (1.1)** | **51.2 (0.8)** | **85.4 (0.3)** | **53.4 (0.5)** | **76.6 (0.1)** |

we employ the Louvain method (Blondel et al., 2008) for data partitioning. For all datasets, the global graph is distributed across 10 clients, with data partitioned into training, validation, and testing sets following a 60%/20%/20% ratio. We select ACM-GNN (Luan et al., 2021) as the shared backbone architecture for all evaluated methods, owing to its demonstrated superiority in handling heterophily. To ensure statistical reliability, experiments are conducted over five independent runs, and we report the mean accuracy (ACC) with standard deviation. For FedGCM, the group size $K$ is set to 4 and the collaboration strength $\eta$ is tuned within the range [0.1, 1.0].

### 4.2. Performance Results

Table 2 presents the comparative results between FedGCM and various competing algorithms. The results indicate that FedGCM consistently achieves state-of-the-art performance, significantly outperforming all baselines across both homophilous and heterophilous benchmarks. Empowered by the ACM-GNN backbone, which is specifically tailored to address graph heterophily, the trained GNN architectures can effectively capture complex feature interactions among heterophilous nodes. This is evidenced by the fact that $FedAvg_{ACM}$ significantly surpasses $FedAvg_{GCN}$ on heterophilous datasets. Among established FGL approaches, frameworks designed to mitigate homophily heterogeneity (such as AdaFGL and FedHERO) exhibit comparable performance on heterophilous datasets, with AdaFGL attaining sub-optimal results on most cases. Furthermore, unlike prior methods that impose substantial local computational complexity or communication overheads, FedGCM innovatively employs server-side gradient surgery to alleviate optimization conflicts. This strategy improves predictive performance without incurring additional client-side overhead, striking a balance between model performance and communication efficiency. A detailed complexity analysis

is provided in Section 4.4.

Additional comparison results, including those under large-scale client settings with 50 and 100 clients, are provided in Appendix D.

### 4.3. Parameter Sensitivity

The sensitivity analysis regarding the collaboration strength $\eta$ and the group size $K$ is illustrated in Figure 3. As depicted in Figure 3(a), optimal performance is typically observed within the range $\eta \in [0.2, 0.8]$, indicating that moderate integration of inter-group information is essential for enhancing the model's generalization performance. Moreover, excessive influence can disrupt local group updates on some datasets, as evidenced by the sharp performance drop on Amazon-ratings at $\eta = 1$.

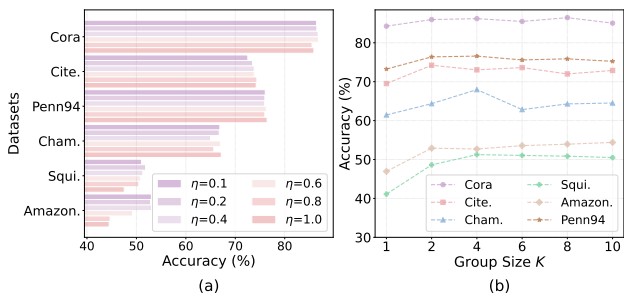

*Figure 3.* Parameter sensitivity analysis. (a) Test accuracy under varying the trade-off hyperparameter $\eta$. (b) Test accuracy with respect to the group size $K$.

Regarding the number of groups $K$, Figure 3(b) highlights a critical trade-off. Setting $K = 1$ reduces the framework to FedAvg, which is inherently hampered by global gradient conflicts. On the other extreme, an excessively large $K$ leads to over-partitioning, which dilutes the collaborative gains within groups. Consequently, $K = 4$ is identified as our configuration, effectively balancing the need for intra-group

*Table 3.* Comparison of spatial-temporal complexity and communication overhead of FGL methods addressing homophily heterogeneity.

| | Methods | Communication Cost | Server Spa. Comp. | Client Spa. Comp. | Server Temp. Comp. | Client Temp. Comp. |
|---|---|---|---|---|---|---|
| Backbone | FedAvg | $\mathcal{O}(MD^2)$ | $\mathcal{O}(M(1+D^2))$ | $\mathcal{O}(N_iD+|\mathcal{E}_i|+D^2)$ | $\mathcal{O}(MD^2)$ | $\mathcal{O}(L(|\mathcal{E}_i|D+N_iD^2))$ |
| Competitors | AdaFGL | $\mathcal{O}(MD^2)$ | $\mathcal{O}(M(1+D^2))$ | $\mathcal{O}(N_iD+|\mathcal{E}_i|+M'D^2)$ | $\mathcal{O}(MD^2)$ | $\mathcal{O}(L(|\mathcal{E}_i|D+N_iD^2)+L'|\mathcal{E}_i|C)$ |
| | FedGSP | $\mathcal{O}(M(D^2+L+D))$ | $\mathcal{O}(ML(D^2+M))$ | $\mathcal{O}(N_iD+|\mathcal{E}_i|+D^2+LN_iD)$ | $\mathcal{O}(MD(M+D+1)+ML)$ | $\mathcal{O}(L(|\mathcal{E}_i|D+N_iD^2))$ |
| | FedHERO | $\mathcal{O}(M(D^2+HD))$ | $\mathcal{O}(M(1+D^2))$ | $\mathcal{O}(N_iD+|\mathcal{E}_i|+K'N_i+M'D^2)$ | $\mathcal{O}(MD^2)$ | $\mathcal{O}(L(|\mathcal{E}_i|D+N_iD^2)+HN_iD)$ |
| | FedSPA | $\mathcal{O}(M'MD^2)$ | $\mathcal{O}(M'M(1+D^2))$ | $\mathcal{O}(N_iD+|\mathcal{E}_i|+M'D^2)$ | $\mathcal{O}(M'MD^2)$ | $\mathcal{O}(M'L(|\mathcal{E}_i|D+N_iD^2))$ |
| Ours | FedGCM | $\mathcal{O}(MD^2)$ | $\mathcal{O}(M(1+D^2)+KD^2)$ | $\mathcal{O}(N_iD+|\mathcal{E}_i|+D^2)$ | $\mathcal{O}(MD^2+K^2D^2)$ | $\mathcal{O}(L(|\mathcal{E}_i|D+N_iD^2))$ |

consistency with inter-group collaboration.

## 4.4. Computational and Communication Overhead

Let $M, M', D, L, K, N_i, |\mathcal{E}_i|, L', K'$ and $H$ denote the number of clients, models, hidden dimension, layers, groups, node/edge counts, label propagation layers, neighbors of $k$-nearest neighbor algorithm and attention heads, respectively. We present the spatial-temporal complexity and communication overhead for methods addressing homophily heterogeneity in Table 3. FedGCM maintains client-side complexity and communication overhead consistent with standard FedAvg. The computational load is mainly shifted to the server, with a one-time $\mathcal{O}(M)$ cost for group index assignment and per-round costs of $\mathcal{O}(MD^2)$ for group aggregation and $\mathcal{O}(K^2D^2)$ for gradient surgery. Although FedGCM introduces extra server-side computation, the overhead is manageable because conflict mitigation is performed at the group level rather than pairwise across clients. Unlike competing methods such as FedGSP, FedHERO, and FedSPA, which incur extra communication costs for auxiliary data or local computational demands, FedGCM enhances global performance while minimizing resource consumption on edge devices. Although FedGCM requires an initial estimation of the high-frequency area, this operation is performed only once before federated training, and the score is treated as a fixed attribute of each client.

*Table 4.* Time and memory comparison on two datasets: PubMed and Amazon-ratings.

| PubMed | Time@10 | Mem-S@10 | Mem-C@10 | Time@50 | Mem-S@50 | Mem-C@50 |
|---|---|---|---|---|---|---|
| FedAvg (backbone) | 38.1 | 232.0 | 304.2 | 173.7 | 148.1 | 161.5 |
| FedGSP | 362.6 | 1413.8 | 838.0 | 3182.4 | 6310.4 | 957.0 |
| AdaFGL | 86.1 | 276.2 | 330.5 | 400.0 | 226.0 | 174.8 |
| FedSPA | 68.7 | 411.6 | 333.7 | 276.3 | 450.3 | 289.9 |
| FedGCM | 41.2 | 252.7 | 308.1 | 258.9 | 169.3 | 164.4 |
| Amazon. | Time@10 | Mem-S@10 | Mem-C@10 | Time@50 | Mem-S@50 | Mem-C@50 |
| FedAvg (backbone) | 42.0 | 294.0 | 380.0 | 195.2 | 98.2 | 114.0 |
| FedGSP | 303.9 | 676.4 | 557.4 | 2164.2 | 3375.2 | 1116.5 |
| AdaFGL | 75.9 | 340.3 | 398.8 | 530.2 | 174.5 | 153.5 |
| FedSPA | 137.8 | 2009.1 | 1783.7 | 275.9 | 378.8 | 241.6 |
| FedGCM | 42.7 | 314.3 | 383.3 | 271.9 | 118.8 | 116.0 |

We further compare runtime (Time), client-side memory (Mem-C), and server-side memory (Mem-S) on PubMed and Amazon-ratings with 10 and 50 clients under the same setting: 100 communication rounds, 5 local epochs, and a 3-layer ACM-GNN with hidden size 64. Client-side mem-

ory is reported as the average total memory across selected clients, while server-side memory is measured during aggregation, with total memory defined as peak GPU reserved memory plus peak CPU RSS increment. As shown in Table 4, FedAvg is the fastest and most memory-efficient due to its single-model client design and simple aggregation in the server side. FedGCM achieves better performance than FedAvg (Table 2) while keeping client-side memory and computation close to the backbone and outperforming multi-channel baselines.

## 4.5. Ablation Study

As illustrated in Figure 4, we examine a variant of FedAvg augmented with our homophily-based grouping and group-oriented broadcasting mechanisms, yet excluding inter-group information exchange. The results demonstrate that this grouping design yields a significant performance improvement over FedAvg with the same GNN backbone. This finding confirms that partitioning clients based on homophily levels encourages intra-group aggregation among clients with more consistent optimization directions, thus mitigating gradient conflicts. Moreover, the additional performance gain achieved by FedGCM over this variant highlights the significance of the inter-group collaboration.

*Table 5.* Ablation study for validating the impact of homophily heterogeneity.

| Methods | Cora | CiteSeer | Chameleon | Squirrel | Amazon. | Penn94 |
|---|---|---|---|---|---|---|
| Isolation | 86.0 (0.9) | 72.4 (0.4) | 66.5 (1.5) | 51.1 (0.8) | 52.6 (0.3) | 75.6 (0.7) |
| w/o projection | 81.6 (1.7) | 73.5 (1.1) | 62.3 (2.1) | 41.6 (2.0) | 39.3 (1.0) | 73.6 (0.7) |

Figure 5 evaluates the contribution of RPGrad by comparing it against three variants: 1) w/o projection: direct summation across groups; 2) w/o group-oriented: uniform broadcast of the global mean; 3) w/ PCGrad: replacing RPGrad with PCGrad. We have the following conclusions:

- The significant degradation observed in the w/o projection variant demonstrates that the directly aggregating heterogeneous gradients leads to directional contradictions, which can hinder global optimization. We additionally compare the Intra-group Only (Isolation) variant with the w/o projection variant in Table 5. The results show that the w/o projection variant performs worse than Isolation on most datasets. This indicates

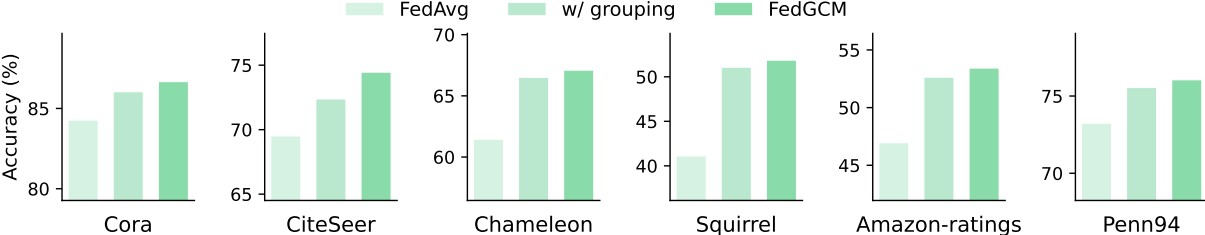

*Figure 4.* Ablation study for validating the effectiveness of the homophily-driven grouping scheme.

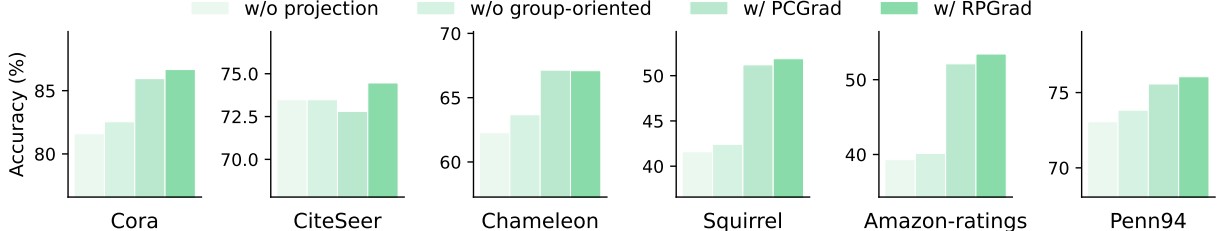

*Figure 5.* Ablation study for validating the effectiveness of the proposed RPGrad and group-oriented broadcast.

that, without an effective projection mechanism to resolve inter-group gradient conflicts, cross-group interactions may become detrimental rather than beneficial to global performance.

- Although conflict-mitigation methods improve performance, RPGrad outperforms PCGrad, further revealing the necessity of integrating cross-group complementary information.

- Additionally, group-oriented broadcasting allows each group to achieve its specific optimization objective, surpassing the uniform broadcast variant.

### 4.6. Convergence Analysis

Figure 6 demonstrates the convergence curves of FedGCM by plotting the mean test accuracy across five random runs on eight datasets. As the communication rounds increase, the model accuracy exhibits a steady gain, eventually reaching a steady-state plateau. This empirical evidence validates both the effectiveness and the convergence of the proposed method. Additional convergence curves of state-of-the-art algorithms are provided in Appendix D.5.

### 5. Conclusion

In this paper, we proposed FedGCM, a novel framework for mitigating optimization conflicts induced by homophily heterogeneity in Federated Graph Learning. By using a homophily-driven grouping strategy with residual projection-based gradient surgery, FedGCM effectively aligned divergent optimization objectives without incurring

*Figure 6.* Convergence curves of FedGCM on eight datasets.

additional client-side computational overhead. Comprehensive evaluations demonstrated that our approach achieved state-of-the-art performance with empirical convergence on both homophilous and heterophilous graphs.

Despite its effectiveness in addressing homophily-heterogeneity-induced optimization conflicts, FedGCM still has several limitations. On the security side, FedGCM is not a dedicated defense method, although RPGrad may partially reduce the influence of malicious updates through projection and adaptive collaboration weights. On the privacy side, FedGCM does not fully eliminate the risk of privacy leakage, since exchanged gradients may still reveal information about local data (Lyu et al., 2020). Strengthening FedGCM with techniques such as differential privacy or secure multiparty computation protocols is an important direction for future work. Moreover, the adaptive selection of $\eta$ remains unresolved. A promising extension is to adjust it according to the degree of gradient alignment across groups.

## Acknowledgments

This work was supported by the National Natural Science Foundation of China (Grant No. 62476133, 62372238) and the Fundamental Research Funds for the Central Universities (Grant No. 11300-312200502507).

## Impact Statement

This paper advances the field of federated graph learning by effectively addressing the challenge of homophily heterogeneity. Our method improves the reliability of collaborative modeling across diverse graph structures, which has positive influence for federated applications such as financial fraud detection and healthcare coordination. Furthermore, by shifting the computational burden to the server, we avoid increasing the overhead on resource-constrained edge devices, promoting the democratization of distributed AI.

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

# A. Proof of Theorem 3.6

We first prove $\langle \widetilde{d}_k^t, \bar{g}_k^t \rangle > 0$, as follows,

$$
\begin{aligned}
\langle \widetilde{d}_k^t, \bar{g}_k^t \rangle &= \langle \bar{g}_k^t + \eta \sum_{j \neq k} \alpha_{kj}^t \hat{g}_j^t, \bar{g}_k^t \rangle \\
&= \langle \bar{g}_k^t, \bar{g}_k^t \rangle + \sum_{j \neq k} \langle \eta \alpha_{kj}^t \hat{g}_j^t, \bar{g}_k^t \rangle \\
&= \|\bar{g}_k^t\|^2 + \sum_{j \neq k} \langle \eta \alpha_{kj}^t \hat{g}_j^t, \bar{g}_k^t \rangle.
\end{aligned}
\tag{15}
$$

Since $\hat{g}_j^t$ is either orthogonal to $\bar{g}_k^t$ or forms an acute angle with it, we have $\langle \hat{g}_j^t, \bar{g}_k^t \rangle \geq 0$. Consequently, the aggregate direction satisfies $\langle \widetilde{d}_k^t, \bar{g}_k^t \rangle > 0$. This indicates that RPGrad preserves positive alignment with the original group-level gradient direction, so updating along $-\widetilde{d}_k^t$ does not introduce a conflicting component with respect to $\bar{g}_k^t$.

**Assumption A.1.** The objective function $\mathcal{L}_k$ is convex, satisfying the following inequality for any model state $\widetilde{w}_k^t$ and the optimal solution $\widetilde{w}_k^*$:

$$
\mathcal{L}_k(\widetilde{w}_k^t) - \mathcal{L}_k(\widetilde{w}_k^*) \leq \langle \nabla \mathcal{L}_k(\widetilde{w}_k^t), \widetilde{w}_k^t - \widetilde{w}_k^* \rangle.
\tag{16}
$$

**Assumption A.2.** The norms of both the gradient $\nabla \mathcal{L}_k(\widetilde{w}_k^t)$ and the update direction $\widetilde{d}_k^t$ are uniformly bounded by a constant $G \geq 0$.

**Lemma A.3.** *There exists a constant $\delta \geq 0$ such that the deviation between the update direction $\widetilde{d}_k^t$ and the gradient $\nabla \mathcal{L}_k(\widetilde{w}_k^t)$ is bounded by:*

$$
\|\widetilde{d}_k^t - \nabla \mathcal{L}_k(\widetilde{w}_k^t)\|_2 \leq \delta.
\tag{17}
$$

*Proof.* We derive the bound by decomposing the deviation into two components using the triangle inequality:

$$
\|\widetilde{d}_k^t - \nabla \mathcal{L}_k(\widetilde{w}_k^t)\|_2 \leq \underbrace{\|\widetilde{d}_k^t - \bar{g}_k^t\|_2}_{\text{Geometric Deviation}} + \underbrace{\|\bar{g}_k^t - \nabla \mathcal{L}_k(\widetilde{w}_k^t)\|_2}_{\text{Estimation Error}}.
\tag{18}
$$

**1. Bounding the Geometric Deviation:** According to the update rule $\widetilde{d}_k^t = \bar{g}_k^t + \eta \sum_{j \neq k} \alpha_{kj}^t \hat{g}_j^t$ and under the bounded group-gradient assumption, we have $\|\bar{g}_j^t\|_2 \leq G$, and the rectified gradient also satisfies $\|\hat{g}_j^t\|_2 \leq G$. Therefore,

$$
\|\widetilde{d}_k^t - \bar{g}_k^t\|_2 \leq \eta \sum_{j \neq k} \alpha_{kj}^t \|\hat{g}_j^t\|_2 \leq \eta \cdot G \cdot \sum_{j \neq k} \alpha_{kj}^t = \eta G.
\tag{19}
$$

**2. Bounding the Estimation Error:** The term $\|\bar{g}_k^t - \nabla \mathcal{L}_k(\widetilde{w}_k^t)\|_2$ represents the discrepancy between the gradient (accumulated over multiple local epochs) and the true gradient. In federated optimization, under standard smoothness and bounded variance assumptions, this error is bounded by a constant, denoted as $\epsilon$ (Reddi et al., 2020; Karimireddy et al., 2020).

Combining both parts, the total deviation is bounded by $\delta = \eta G + \epsilon$. Since $\eta, G$, and $\epsilon$ are constants, $\delta$ is a constant. $\square$

Based on the above Assumptions A.1-A.3, the following theorem holds:

**Theorem A.4.** *Let $\widetilde{w}_k^*$ be an optimal solution for the $k$-th group. For any positive sequence of step sizes $\{\tau^t\}_{t=1}^T$, the minimum optimization gap after $T$ rounds is bounded by:*

$$
\min_{t=1,\ldots,T} \left( \mathcal{L}_k(\widetilde{w}_k^t) - \mathcal{L}_k(\widetilde{w}_k^*) \right) \leq \frac{E^2 + G^2 \sum_{t=1}^T (\tau^t)^2}{2 \sum_{t=1}^T \tau^t} + \delta R,
\tag{20}
$$

*where $E = \|\widetilde{w}_k^1 - \widetilde{w}_k^*\|_2$ is the distance between the initial parameters and the optimal solution and $R$ is the radius of the parameter search space, i.e., $\|\widetilde{w}_k^t - \widetilde{w}_k^*\|_2 \leq R$.*

*Proof.* Consider the Euclidean distance between the model $\widetilde{w}_k^{t+1}$ and the optimal solution $\widetilde{w}_k^*$. Based on the update rule $\widetilde{w}_k^{t+1} = \widetilde{w}_k^t - \tau^t \widetilde{d}_k^t$, we have:

$$\begin{aligned}
\|\widetilde{w}_k^{t+1} - \widetilde{w}_k^*\|_2^2 &= \|\widetilde{w}_k^t - \tau^t \widetilde{d}_k^t - \widetilde{w}_k^*\|_2^2 \\
&= \|\widetilde{w}_k^t - \widetilde{w}_k^*\|_2^2 - 2\tau^t \langle \widetilde{d}_k^t, \widetilde{w}_k^t - \widetilde{w}_k^* \rangle + (\tau^t)^2 \|\widetilde{d}_k^t\|_2^2.
\end{aligned} \tag{21}$$

We analyze the inner product term by introducing the true gradient $\nabla \mathcal{L}_k(\widetilde{w}_k^t)$. The term is decomposed as:

$$-\langle \widetilde{d}_k^t, \widetilde{w}_k^t - \widetilde{w}_k^* \rangle = -\langle \nabla \mathcal{L}_k(\widetilde{w}_k^t), \widetilde{w}_k^t - \widetilde{w}_k^* \rangle + \langle \nabla \mathcal{L}_k(\widetilde{w}_k^t) - \widetilde{d}_k^t, \widetilde{w}_k^t - \widetilde{w}_k^* \rangle. \tag{22}$$

For the first term, applying the convexity assumption (Assumption A.1) yields:

$$-\langle \nabla \mathcal{L}_k(\widetilde{w}_k^t), \widetilde{w}_k^t - \widetilde{w}_k^* \rangle \leq -\Big( \mathcal{L}_k(\widetilde{w}_k^t) - \mathcal{L}_k(\widetilde{w}_k^*) \Big). \tag{23}$$

For the second term, we apply the Cauchy-Schwarz inequality and Assumption A.3, and assume the parameter search space is limited to a radius of $R$, i.e., $\|\widetilde{w}_k^t - \widetilde{w}_k^*\|_2 \leq R$:

$$\langle \nabla \mathcal{L}_k(\widetilde{w}_k^t) - \widetilde{d}_k^t, \widetilde{w}_k^t - \widetilde{w}_k^* \rangle \leq \|\nabla \mathcal{L}_k(\widetilde{w}_k^t) - \widetilde{d}_k^t\|_2 \cdot \|\widetilde{w}_k^t - \widetilde{w}_k^*\|_2 \leq \delta R. \tag{24}$$

Substituting these results back into Eqn. (21) and using Assumption A.2:

$$\|\widetilde{w}_k^{t+1} - \widetilde{w}_k^*\|_2^2 \leq \|\widetilde{w}_k^t - \widetilde{w}_k^*\|_2^2 - 2\tau^t \Big( \mathcal{L}_k(\widetilde{w}_k^t) - \mathcal{L}_k(\widetilde{w}_k^*) \Big) + 2\tau^t \delta R + (\tau^t)^2 G^2. \tag{25}$$

Summing over $t = 1$ to $T$ and using the telescoping sum, we obtain

$$2\sum_{t=1}^{T} \tau^t \Big( \mathcal{L}_k(\widetilde{w}_k^t) - \mathcal{L}_k(\widetilde{w}_k^*) \Big) \leq \|\widetilde{w}_k^1 - \widetilde{w}_k^*\|_2^2 + 2\delta R \sum_{t=1}^{T} \tau^t + G^2 \sum_{t=1}^{T} (\tau^t)^2. \tag{26}$$

Dividing both sides by $2\sum_{t=1}^{T} \tau^t$ and denoting $E = \|\widetilde{w}_k^1 - \widetilde{w}_k^*\|_2$, we have

$$\frac{\sum_{t=1}^{T} \tau^t \Big( \mathcal{L}_k(\widetilde{w}_k^t) - \mathcal{L}_k(\widetilde{w}_k^*) \Big)}{\sum_{t=1}^{T} \tau^t} \leq \frac{E^2 + G^2 \sum_{t=1}^{T} (\tau^t)^2}{2\sum_{t=1}^{T} \tau^t} + \delta R. \tag{27}$$

Since the minimum optimization gap is no larger than the weighted average optimization gap, we obtain

$$\min_{t=1,\dots,T} \Big( \mathcal{L}_k(\widetilde{w}_k^t) - \mathcal{L}_k(\widetilde{w}_k^*) \Big) \leq \frac{E^2 + G^2 \sum_{t=1}^{T} (\tau^t)^2}{2\sum_{t=1}^{T} \tau^t} + \delta R. \tag{28}$$

When a diminishing step size strategy is used, such as $\tau^t \propto \frac{1}{\sqrt{t}}$, the numerator of the bound in Eqn. (28) grows logarithmically ($\sum (\tau^t)^2 \propto \ln T$), while the denominator grows at a faster rate ($\sum \tau^t \propto \sqrt{T}$). Consequently, the first term vanishes, and the optimization gap can converge to $\delta R$, which implies convergence to a neighborhood of size $\delta R$ around the optimum, that is,

$$\lim_{T \to \infty} \Big( \min_{t=1,\dots,T} \big( \mathcal{L}_k(\widetilde{w}_k^t) - \mathcal{L}_k(\widetilde{w}_k^*) \big) \Big) \leq \delta R. \tag{29}$$

## B. FedGCM vs. CFL

FedGCM maintains a globally shared knowledge space after grouping, rather than learning isolated cluster-specific models as in standard Clustering Federated Learning (CFL). The main differences are as follows: 1) Standard CFL methods usually partition clients into disjoint clusters and train separate cluster-specific models with little or no cross-cluster interaction to mitigate Non-IID interference (Ghosh et al., 2020), whereas FedGCM explicitly preserves cross-group interaction through RPGrad in Eqn. (9); 2) In CFL, clustering is the final objective for personalization, while in FedGCM, grouping is introduced to better avoid optimization conflicts under homophily heterogeneity.

## C. Datasets

We perform node classification tasks on both homophilous and heterophilous graph datasets. Their statistics are provided in Table 6, and descriptions are as following

- **Cora, CiteSeer and PubMed** are three popular citation networks in machine learning. Each paper is denoted as a node and edges indicate the citation relationships. Every paper either cites or is cited by at least one other paper.

- **Physics** is derived from the Microsoft Academic Graph and is used in the KDD Cup 2016 challenge, where nodes are authors and edges denote co-authorship relationships.

- **Chameleon and Squirrel** datasets are two page-page networks extracted from Wikipedia (). In these graphs, nodes represent individual web pages, while edges signify mutual hyperlinks between them. Pages are categorized into one of five levels based on their average monthly traffic volume.

- **Minesweeper** is a dataset inspired by the Minesweeper game. This dataset is a regular 100x100 grid, where nodes represent cells connected to their eight neighbors, with 20% of nodes randomly assigned as mines. The task is to predict mine locations based on one-hot encoded features of neighboring mine counts, though 50% of these features are intentionally masked to increase difficulty.

- **Amazon-ratings** comes from the Amazon product co-purchasing network metadata dataset from SNAP (Jure, 2014), where nodes are products and edges indicate products that are frequently bought together.

- **Penn94** is a social network, which is used to predict reported gender, certain account labels, or use of explicit content on user accounts.

*Table 6.* Statistics of four homophilous and five heterophilous graphs.

|  | Datasets | #Nodes | #Features | #Edges | #Classes | #Homo. |
|---|---|---|---|---|---|---|
| Homophilous Datasets | Cora | 2,708 | 1,433 | 5,429 | 7 | 0.810 |
|  | CiteSeer | 3,327 | 3,703 | 4,732 | 6 | 0.736 |
|  | PubMed | 19,717 | 500 | 44,338 | 3 | 0.802 |
|  | Physics | 34,493 | 8,415 | 247,962 | 5 | 0.931 |
| Heterophilous Datasets | Chameleon | 2,277 | 2,325 | 36,101 | 5 | 0.234 |
|  | Squirrel | 5,201 | 2,089 | 216,933 | 5 | 0.223 |
|  | Minesweeper | 10,000 | 7 | 39,402 | 2 | 0.683 |
|  | Amazon-ratings | 24,492 | 300 | 93,050 | 5 | 0.380 |
|  | Penn94 | 41,554 | 4,814 | 1,362,229 | 2 | 0.470 |

## D. More Experiments

### D.1. Effectiveness of Louvain Partitioning

To verify the effectiveness of Louvain partitioning in simulating cross-client homophily heterogeneity, we report the actual heterophily levels of ten clients on the Chameleon dataset. As shown in Table 7, the heterophily ratios vary across clients, indicating that the partitioning strategy can effectively simulate the desired cross-client heterogeneity scenario.

*Table 7.* Heterophily levels across 10 clients on the Chameleon dataset.

| Dataset | C0 | C1 | C2 | C3 | C4 | C5 | C6 | C7 | C8 | C9 |
|---|---|---|---|---|---|---|---|---|---|---|
| Chameleon | 0.716 | 0.794 | 0.765 | 0.765 | 0.571 | 0.724 | 0.728 | 0.491 | 0.724 | 0.661 |

## D.2. Performance Comparison on Large-scale Clients

We further scale the number of clients to 50 and 100 on relatively large datasets, including PubMed, Physics, Minesweeper, and Penn94. For Penn94, we use Metis (Karypis & Kumar, 1998) instead of Louvain in the 50- and 100-client settings, since Louvain produces empty clients at larger scales. The results in Table 8 show that FedGCM still achieves the best performance in most cases, demonstrating its effectiveness and practicality beyond the 10-client setting.

*Table 8.* Performance comparison on four Datasets with large-scale clients.

| Method | PubMed@50 | PubMed@100 | Physics@50 | Physics@100 | Penn94@50 | Penn94@100 | Minesweeper@50 | Minesweeper@100 |
|---|---|---|---|---|---|---|---|---|
| FedAvg | 87.8 | 87.9 | 95.8 | 95.6 | 69.3 | 68.0 | 80.2 | 80.8 |
| FedNova | 85.2 | 84.6 | 94.1 | 93.9 | 67.2 | 68.2 | 79.4 | 81.4 |
| FedPUB | 87.2 | 86.9 | 95.6 | 94.8 | 71.5 | 69.2 | 80.3 | 80.6 |
| FedGTA | 85.8 | 85.0 | 94.2 | 94.5 | 70.0 | 67.4 | 80.3 | 81.0 |
| AdaFGL | 87.8 | 88.4 | 96.0 | 95.6 | 74.1 | 74.0 | 81.6 | 82.1 |
| FedTAD | 88.1 | **89.4** | 96.2 | 95.1 | 70.2 | 69.2 | 80.3 | 80.5 |
| FedHERO | 82.8 | 84.5 | 95.4 | 95.2 | OoM | OoM | 82.6 | 80.9 |
| FedSPA | 86.7 | 87.1 | 95.1 | 95.3 | **74.7** | **74.2** | 82.4 | 81.3 |
| Ours | **88.6** | 88.8 | **96.2** | **95.9** | 74.2 | 74.0 | **84.3** | **82.6** |

## D.3. Performance Analysis under the GCN Backbone

We add the comparison results under the GCN backbone, as shown in Table 9. After replacing ACM-GNN with GCN, most methods show clear performance drops, especially on heterophilous datasets, due to GCN's weaker heterophily modeling ability. Nevertheless, FedGCM still achieves the best results on most datasets, showing that its effectiveness does not solely depend on a specific strong backbone.

*Table 9.* GCN-Based performance comparison on six datasets

| Methods | Cora | CiteSeer | Physics | Chameleon | Squirrel | Amazon-ratings |
|---|---|---|---|---|---|---|
| FedAvg | 84.7 (0.5) | 71.4 (0.9) | 86.7 (0.5) | 38.7 (0.9) | 26.6 (1.1) | 47.7 (0.5) |
| FedNova | 83.9 (0.5) | 74.1 (0.5) | 94.6 (0.2) | 36.9 (1.4) | 24.2 (1.7) | 46.0 (0.2) |
| FedPUB | 83.9 (1.2) | 72.7 (2.4) | 95.8 (0.2) | 58.8 (1.8) | 42.6 (1.5) | 47.9 (0.4) |
| FedGTA | 83.3 (0.3) | 74.0 (1.0) | 94.4 (0.1) | 38.0 (1.3) | 26.1 (1.7) | 47.4 (0.2) |
| AdaFGL | 83.2 (0.2) | 72.5 (0.9) | 94.7 (0.2) | 48.7 (0.2) | 37.6 (0.6) | **52.4 (0.4)** |
| FedGSP | 83.5 (0.9) | 68.5 (0.4) | 95.3 (0.3) | 55.1 (0.2) | 37.8 (0.1) | 46.4 (0.1) |
| FedHERO | 82.3 (1.0) | 70.1 (1.3) | 95.9 (0.2) | 57.7 (1.0) | 36.8 (2.4) | 50.7 (0.2) |
| FedGCM | **85.4 (0.1)** | **74.2 (0.4)** | **96.2 (0.1)** | **63.2 (1.6)** | **44.7 (0.6)** | 50.1 (0.4) |

## D.4. Performance Comparison under Different Ratios

We conducted additional experiments to further validate the effectiveness of FedGCM. Figure 7 reports the performance curves of different methods under various training label ratios, including 0.2, 0.4, and 0.8. The results show that FedGCM achieves competitive or superior performance in most cases across both low- and high-label-ratio settings.

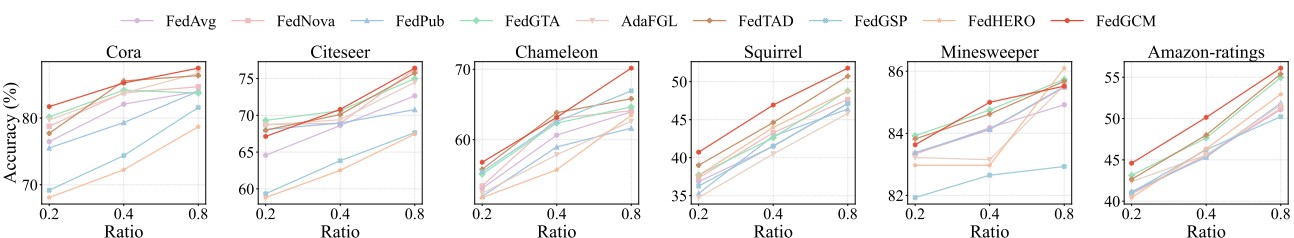

*Figure 7.* Performance curves on 6 datasets with training ratios of 0.2, 0.4 and 0.8.

## D.5. Convergence Analysis

We compare the convergence curves between FedGCM and other SOTA algorithms on several datasets, as shown in Figure 8. The results show that FedGCM converges stably and achieves stronger performance than the compared baselines.

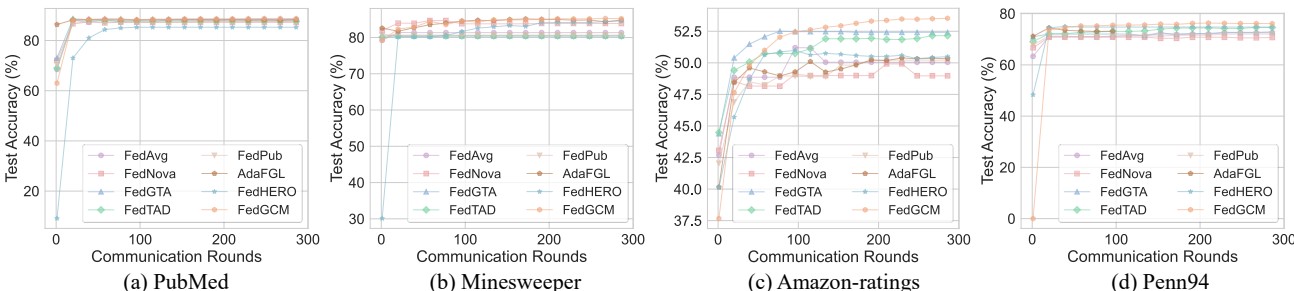

(a) PubMed      (b) Minesweeper      (c) Amazon-ratings      (d) Penn94

*Figure 8.* Average test ACC curves of FedGCM and seven competing methods over communication rounds, with the number of clients set to 10.

