# OpenReview forum: "Homophily-Heterogeneity Gradient Surgery for Federated Graph Learning"
_ICML.cc/2026/Conference — ICML 2026 regular_

### Official Review · Reviewer_eGC5 · 2026-03-04

**Soundness:** 3
**Presentation:** 4
**Significance:** 3
**Originality:** 3
**Overall Recommendation:** 4
**Confidence:** 4

**Summary:**

To address optimization conflicts induced by homophily heterogeneity in FGL, many advanced methods are proposed. Despite these advancements, they are multi-channel architectures, maintaining multiple model branches double the parameter count and exacerbates memory overhead. Furthermore, they lack formal theoretical convergence guarantees, making it challenging to ensure model stability during the collaborative optimization on distributed data. Based on the above limitations, FedGCM aims to resolve optimization conflicts through inter-group gradient surgery while maintaining a streamlined architecture on the client side.

**Compliance With Llm Reviewing Policy:**

Affirmed.

**Key Questions For Authors:**

See the Weaknesses.

**Limitations:**

No. While FedGCM maintains a single model on the client side, the gradient surgery performed in each round significantly increases the computational burden on the server. Furthermore, there is still a risk of the privacy leakage associated with sharing information including gradients and homophily levels.

**Strengths And Weaknesses:**

Strengths.
1.The introduction section is well written, clarifying the existing problems and related solutions.
2.The proposed model is innovative, which uses designed gradient surgery to avoid the burdens derived from classic multi-channel frameworks, and obtains SOTA performance on nine datasets.
3.Rigorous theoretical analysis has verified the convergence of the proposed method.
Weaknesses.
1.Figure 2 is unclear. It is suggested that the authors revise this figure to more explicitly present the various submodules and their relationships.
2.While FedGCM maintains a single model on the client side, it simply shifts a massive computational bottleneck to the server, where the gradient surgery requires $\mathcal{O}(K^2 D^2)$ time complexity.
3.Is the configuration $K=4$ (L416) optimized per dataset using validation sets, or is it a uniform setting used throughout the experiments?
4.Insufficient experiments. Although Section 4 covers the performance comparison, ablation study, parameter sensitivity etc., these only are a basic assessment. To further demonstrate its effectiveness, the authors should consider adding results for scenarios involving large-scale client settings (e.g., 50, 100) or different label ratios.
5.The convergence analysis in Section 4.6 only includes the proposed model. Convergence curves of other baseline methods should be incorporated to provide a more comprehensive evaluation.
6.Nine datasets are used for general evaluation, but six datasets are included in the ablation study, and the results should be extended to all nine datasets.

---

> ### Author Rebuttal · Authors · 2026-03-30
>
> Thank you for the valuable suggestions.
>
> >Q1&Q5. Clearer figure and convergence curves of baselines
>
> We have revised Figure 2 to more clearly present the framework modules and their relationships (https://anonymous.4open.science/r/figures-90CF/framework.pdf), and we have also added convergence comparisons with representative baselines on several datasets (https://anonymous.4open.science/r/figures-90CF/alg_conver.pdf). The results show that FedGCM converges stably and achieves stronger performance than the compared baselines.
>
> >Q2. Server-side computational complexity of FedGCM
>
> We clarify that FedGCM does introduce extra server-side computation, **but the overhead remains manageable**. Because Eq. (9) is computed at the group level, the interaction cost depends on $K$ rather than exhaustive client-to-client comparisons. **Detailed runtime and memory comparisons are provided in our response to #Reviewer-5cgz**. Unlike multi-channel methods, FedGCM does not significantly increase client-side memory, making the extra server overhead acceptable in practice.
>
> >Q3. Choice of the hyperparameter $K$
>
> We use **a uniform setting of $K=4$ for all datasets in the main experiments, as stated in Section 4.1**, rather than tuning it per dataset using validation sets. We also **analyze $K  \in \lbrace 1,2,4,6,8,10 \rbrace$ in Figure 3(b)**. Although a few datasets perform slightly better with other values, $K=4$ provides a stable and effective default across datasets.
>
> >Q4. Insufficient experiments
>
> Following your comments, we have conducted additional experiments to further validate the effectiveness of FedGCM.
> First, **the following tables report the performance of all methods under different training label ratios (0.2 and 0.8)**. The results indicate that FedGCM consistently achieves superior performance in most cases across both low- and high-label-ratio settings.
> Secondly, we further evaluate large-scale client settings. Please refer to our response to **#Reviewer HUtS-Q6** for detailed results and discussion.
> | Ratio=0.2 | Physics | PubMed | Amazon. | Minesweeper | Penn94 |
> |--------|------|-----|-----|-----|-----|
> | FedAvg | 95.6 (0.1) | 85.3 (0.5) | 41.0 (0.4) | 83.3 (0.4) | 68.6 (0.7) |
> | FedNova| 95.7 (0.1) | 85.3 (0.3) | 40.9 (0.5) | 83.3 (0.1) | 68.6 (0.4) |
> | FedPub | 95.7 (0.1) | 84.8 (0.2) | 41.2 (0.5) | 83.4 (0.2) | 69.4 (0.5) |
> | FedGTA | 95.9 (0.0) | 86.2 (0.3) | 43.1 (0.2) | 83.5 (0.2) | 68.7 (0.9) |
> | AdaFGL | 95.4 (0.1) | 85.7 (0.2) | 42.4 (0.4) | 83.2 (0.4) | 73.8 (0.3) |
> | FedTAD | 95.6 (0.0) | **86.4 (0.1)** | 42.7 (0.4) | 83.1 (0.4) | 70.3 (1.2) |
> | FedHERO| 94.1 (0.1) | 82.5 (0.4) | 40.4 (0.1) | 83.0 (0.8) | 70.0 (0.3) |
> | FedSPA | 95.4 (0.0) | 84.4 (0.2) | 41.4 (0.1) | 82.7 (0.4) | 73.0 (0.2) |
> | Ours   | **96.0 (0.1)** | 86.1 (0.2) | **44.6 (0.2)** | **83.6 (0.1)** | **75.6 (0.2)** |
>
> | Ratio=0.8 | Physics | PubMed | Amazon. | Minesweeper | Penn94 |
> |--------|------|-----|-----|-----|-----|
> | FedAvg | 96.6 (0.0) | 88.0 (0.5) | 51.1 (0.6) | 84.9 (0.4) | 71.6 (1.0) |
> | FedNova| 96.5 (0.1) | 86.9 (0.4) | 51.2 (0.3) | 85.5 (0.2) | 70.8 (0.7) |
> | FedPub | **96.9 (0.0)** | 88.2 (0.1) | 51.9 (0.3) | 85.5 (0.5) | 72.8 (0.7) |
> | FedGTA | 94.8 (0.0) | 88.6 (0.4) | 55.0 (0.5) | 85.7 (0.8) | 72.7 (0.3) |
> | AdaFGL | 96.2 (0.0) | 88.8 (0.5) | 51.6 (0.4) | 85.5 (0.6) | 75.2 (0.6) |
> | FedTAD | 96.7 (0.1) | 88.9 (0.2) | 55.4 (0.5) | 85.7 (0.4) | 74.9 (1.6) |
> | FedHERO| 96.2 (0.3) | 85.9 (0.5) | 52.9 (0.7) | 86.1 (1.2) | **79.3 (0.1)** |
> | FedSPA | 96.4 (0.2) | 87.7 (0.2) | 42.5 (0.6) | 85.4 (0.8) | 74.5 (0.6) |
> | Ours   | 96.4 (0.1) | **89.5 (0.1)** | **56.1 (0.2)** | **86.5 (0.1)** | 76.8 (0.2) |
>
> >Q6. More ablation study
>
> We agree that ablation results on all nine datasets would provide a more complete validation. Accordingly, we have added the ablation results on the remaining three datasets, as shown in the table below. The conclusions remain consistent with those in the main paper, and FedGCM still achieves the best performance across these datasets.
> | Method               | Pubmed       | Physics      | Minesweeper  |
> |--------------------|-------------|-------------|-------------|
> | w/o projection      | 82.1 (0.9)  | 95.4 (0.0)  | 82.4 (1.7)  |
> | w/o group-oriented  | 82.0 (0.8)  | 94.8 (0.0)  | 83.0 (1.1)  |
> | w/ PCGrad           | 88.5 (0.1)  | 95.7 (0.1)  | 85.2 (0.3)  |
> | w/ grouping         | 88.3 (0.1)  | 95.9 (0.1)  | 85.2 (0.4)  |
> | **FedGCM**          | **88.5 (0.2)** | **96.1 (0.1)** | **85.4 (0.3)** |
>
> >Q7. Limitations
>
> Although FedGCM incurs additional server-side computation, this overhead remains manageable, as conflict mitigation is conducted at the group level rather than across all client pairs. As detailed in Section 4.5, the server temp. comp. increases from $\mathcal{O}(MD^2)$ in FedAvg to $\mathcal{O}(MD^2+K^2D^2)$ in FedGCM. We will add a Discussion section to clarify the limitation of FedGCM.  Please refer to our response to **#Reviewer FEts-Q6**.

---

> > ### Author Rebuttal · Reviewer_eGC5 · 2026-04-03
> >
> > Thank you for the author's responses. Rebuttal has resolved my concerns, and I maintain a positive score.

---

> > > ### Author Response · Authors · 2026-04-03
> > >
> > > Dear Reviewer eGC5,
> > >
> > > Thank you very much for your timely follow-up and positive reassessment of the paper. Your constructive comments and support are very valuable to us.
> > >
> > > We will carefully revise the manuscript in accordance with your suggestions for further improving the clarity and completeness of the paper.
> > >
> > > Thank you again for your time.
> > >
> > > Sincerely,
> > >
> > > Authors

---

### Official Review · Reviewer_FEts · 2026-03-04

**Soundness:** 3
**Presentation:** 4
**Significance:** 3
**Originality:** 4
**Overall Recommendation:** 5
**Confidence:** 5

**Summary:**

This paper addresses a challenge in Federated Graph Learning (FGL) homophily heterogeneity. When subgraphs across different clients exhibit significantly different homophily levels, standard aggregation methods like FedAvg suffer from optimization conflicts, leading to poor model performance. To address this problem, the paper proposes homophily-driven grouping and gradient surgery based on residual projection. Moreover, the paper provides a convergence guarantee.

**Compliance With Llm Reviewing Policy:**

Affirmed.

**Final Justification:**

After reading the author's rebuttal and considering other reviewers' comments, the rebuttal has addressed my concerns. The added experiments further verify the effectiveness of the proposed method. I decide to increase my score.

**Key Questions For Authors:**

1. Why does homophily heterogeneity reduce global performance? Please provide empirical evidence.
2. A clearer definition of "residual projection" needs to be provided.
3. Can the adopted Louvain partition effectively simulate the target scenario?

**Limitations:**

No. This proposed FedGCM requires the upload of gradients as shared information, which raises concerns of privacy leakage. Including a discussion section is important, which aims to explore how the model could be enhanced by integrating advanced privacy-preserving mechanisms as part of future work.

**Strengths And Weaknesses:**

This paper proposes a novel FGL framework FedGCM to address the challenge of homophily heterogeneity across clients. The manuscript is well-structured with a clear logical flow, supported by theoretical guarantees and experiments that demonstrate its superiority.

However, I have several concerns and questions:
1. Authors claimed that homophily heterogeneity degrades global performance, however, there are no experimental results to support this.
2. In Algorithm 1, the index starts from 0 at line 9 but starts to 1 at line 15. The subscripts need to be uniform.
3. Could the authors provide a clearer definition of “residual projection”? It would be helpful to elaborate on this concept and specify which component of the proposed framework it corresponds to.
4. The paper addressed the issue of homophily heterogeneity. However, it is unclear whether the adopted Louvain partition can effectively simulate the target scenario where homophily levels (or high-frequency area) vary significantly across different clients.
5. The author reported the results of FedAvg using GCN as the backbone. How do other compared algorithms perform under the same architecture? And does the proposed model maintain its superiority under this backbone?

---

> ### Author Rebuttal · Authors · 2026-03-29
>
> Thank you for raising these critical concerns.
>
> >Q1(KQ1). Empirical evidence for the impact of homophily heterogeneity
>
> First, in Figure 5, the **w/o projection variant, which directly combines gradients across groups, performs consistently worse than FedGCM, indicating that such inter-group conflicts can harm global performance**. Second, as shown in the table below, we further **compare the Intra-group Only (Isolation) variant with the w/o projection variant**.  The results show that the w/o projection variant performs worse than the Isolation variant on most datasets. This suggests that, without properly handling inter-group conflicts, interactions across groups may instead impair global performance, further supporting our claim that optimization conflicts caused by homophily heterogeneity can degrade global performance.
> | Datasets    | Cora         | Citeseer     | Chameleon    | Squirrel     | Amazon. | Penn94     |
> |-----------|-------------|--------------|--------------|--------------|----------------|--------------|
> | Isolation | 86.0 (0.9) | 72.4 (0.4) | 66.5 (1.5) | 51.1 (0.8) | 52.6 (0.3)   | 75.6 (0.7) |
> | w/o projection | 81.6 (1.7) | 73.5 (1.1) | 62.3 (2.1) | 41.6 (2.0) | 39.3 (1.0)   | 73.6 (0.7) |
>
> >Q2. Some typos
>
>  Thank you for pointing this out. We have corrected the notation from $\lbrace \widetilde{w}\_k \rbrace_{k=1}^{K}$ to $\lbrace \widetilde{w}\_k \rbrace_{k=0}^{K-1}$.
>
> >Q3(KQ2). Definition of residual projection
>
> Residual projection refers to the gradient component from an auxiliary group that is **used for collaboration with the target group**. Specifically, if the auxiliary gradient does not conflict with the target gradient, we directly use the original auxiliary gradient; otherwise, we remove its conflicting projection on the target gradient and retain only the residual component on the normal plane (Eq. (9)).
>
> >Q4(KQ3). Effectiveness of Louvain Partitioning
>
> To verify the effectiveness of Louvain partitioning for simulating cross-client homophily heterogeneity, we report the **heterophily levels of 10 clients on Chameleon. As shown in the table below, the heterophily levels differ across clients**, suggesting that the partitioning effectively simulates the target scenario.
> | Dataset   | C0   | C1   | C2   | C3   | C4   | C5   | C6   | C7   | C8   | C9   |
> |-----------|------|------|------|------|------|------|------|------|------|----|
> | Chameleon | 0.716| 0.794 | 0.765 | 0.765 | 0.571 | 0.724 | 0.728 | 0.491 | 0.724 | 0.661 |
>
> >Q5. Results on the GCN backbone
>
> We have added the comparison results under the GCN backbone, as shown in the table below. After replacing ACM-GNN with GCN, **most methods show clear performance drops**, especially on heterophilous datasets, due to GCN’s weaker heterophily modeling ability. Nevertheless, **FedGCM still achieves the best results on most datasets**, showing that its effectiveness does not solely depend on a specific strong backbone.
> | Methods  | Cora        | Citeseer   | Physics     | Chameleon   | Squirrel    | Amazon|
> |----------|------------|------------|------------|------------|------------|---------------|
> | FedAvg   | 84.7 (0.5) | 71.4 (0.9) | 86.7 (0.5) | 38.7 (0.9) | 26.6 (1.1) | 47.7 (0.5) |
> | FedNova  | 83.9 (0.5) | 74.1 (0.5) | 94.6 (0.2) | 36.9 (1.4) | 24.2 (1.7) | 46.0 (0.2) |
> | FedPub   | 83.9 (1.2) | 72.7 (2.4) | 95.8 (0.2) | 58.8 (1.8) | 42.6 (1.5) | 47.9 (0.4) |
> | FedGTA   | 83.3 (0.3) | 74.0 (1.0) | 94.4 (0.1) | 38.0 (1.3) | 26.1 (1.7) | 47.4 (0.2) |
> | AdaFGL   | 83.2 (0.2) | 72.5 (0.9) | 94.7 (0.2) | 48.7 (0.2) | 37.6 (0.6) | **52.4 (0.4)** |
> | FedGSP   | 83.5 (0.9) | 68.5 (0.4) | 95.3 (0.3) | 55.1 (0.2) | 37.8 (0.1) | 46.4 (0.1) |
> | FedHERO  | 82.3 (1.0) | 70.1 (1.3) | 95.9 (0.2) | 57.7 (1.0) | 36.8 (2.4) | 50.7 (0.2) |
> | **FedGCM**  | **85.4 (0.1)** | **74.2 (0.4)** | **96.2 (0.1)** | **63.2 (1.6)** | **44.7 (0.6)** | 50.1 (0.4) |
>
> >Q6. Limitations
>
> We will add a dedicated Discussion section to clarify the limitation of FedGCM: Although FedGCM effectively addresses homophily-heterogeneity-induced optimization conflicts in FGL, **it still has several limitations**. On the security side, FedGCM is not a dedicated defense method, although RPGrad may partially reduce the impact of malicious updates through projection and adaptive collaboration weights. On the privacy side, while FedGCM inherits the decentralized FGL advantage of exchanging gradients rather than sharing graph data, it does not fully eliminate the risk of privacy leakage, since gradients may still reveal information about local data [1]. Strengthening FedGCM with techniques such as differential privacy or secure multi-party computation protocols is an important direction for future work. Moreover, the adaptive learning of $\eta$ remains unresolved; a possible extension is to adjust it according to the degree of gradient alignment across groups.
>
> [1] Lyu, L., et al. Threats to federated learning: A survey. 2020, arXiv preprint arXiv:2003.02133.

---

> > ### Author Rebuttal · Reviewer_FEts · 2026-04-03
> >
> > Thank you for your detailed response. The rebuttal has addressed my concerns. The added experiments further verify the effectiveness of the proposed method. I hope these points will be incorporated into the final version. Based on the rebuttal and the overall responses, I decide to increase my score to accept.

---

> > > ### Author Response · Authors · 2026-04-03
> > >
> > > Dear Reviewer FEts,
> > >
> > > Thank you very much for your timely response and for your recognition of our rebuttal. We sincerely appreciate your encouraging feedback and your decision to raise the score.
> > >
> > > Your suggestions are very valuable for improving the paper, and we will carefully incorporate them in final version.
> > >
> > > Thank you once again for your time and valuable feedback.
> > >
> > > Sincerely,
> > >
> > > Authors

---

### Official Review · Reviewer_HUtS · 2026-03-13

**Soundness:** 3
**Presentation:** 3
**Significance:** 3
**Originality:** 3
**Overall Recommendation:** 5
**Confidence:** 4

**Summary:**

This paper proposes FedGCM, a novel federated graph learning framework designed to resolve optimization conflicts caused by homophily heterogeneity across subgraphs. To circumvent the high computational and communication overhead imposed on edge devices by existing multi-channel architectures, FedGCM shifts the conflict resolution mechanism to the server. Specifically, it groups clients in an unsupervised manner using the high-frequency area of graph signals as a proxy. Subsequently, a residual projection-based gradient surgery (RPGrad) is applied on the server to eliminate inter-group conflicts while integrating synergistic knowledge. Finally, the server broadcasts the refined parameters in a group-oriented fashion, enabling clients to maintain a streamlined single-model architecture. The method demonstrates strong performance on both homophilous and heterophilous datasets and provides theoretical convergence guarantees.

**Compliance With Llm Reviewing Policy:**

Affirmed.

**Key Questions For Authors:**

1.Necessity and Superiority of Inter-Group Collaboration. Since the framework has already identified that certain groups have conflicting gradient directions, why not simply employ "Group-wise Isolation"?

2.Effectiveness of the $S_i$ Proxy. The paper simplifies graph heterogeneity into a single scalar $S_i$. In reality, two subgraphs with the same $S_i$ might produce diametrically opposed gradients.

3.Innovativeness of FedGCM. Discuss the difference between CFL and FedGCM. Additionally, is its performance gain attributed to the use of the powerful heterophily-aware backbone?

**Limitations:**

No.  In practical scenarios, the presence of malicious clients must be considered. Such an adversary could infiltrate a specific group by spoofing its homophily features and subsequently manipulate gradient directions to interfere with the server-side projection operations. A discussion on FedGCM’s security and privacy needs to be provided.

**Strengths And Weaknesses:**

# Strengths

1. Unlike prior multi-branch FGL frameworks (e.g., FedSPA or FedHERO) that double the parameter count and exacerbate memory overhead, FedGCM elegantly shifts the computational burden to the central server.

2. Acknowledging the strict limitations of label inaccessibility in semi-supervised settings, the paper cleverly utilizes the high-frequency area of the graph spectrum, which is inversely correlated with the homophily level, as an unsupervised proxy.

3. Most existing methods are predominantly heuristic-driven. In contrast, this paper provides a formal theoretical convergence analysis for the proposed RPGrad mechanism.

# Weaknesses
1. In equation (9), when the gradients of two groups conflict, i.e., $cos(g_k^t, g_j^t) < 0$,
RPGrad performs orthogonal projection and adds a collaboration term $\eta \sum_{j \neq k} \alpha_{kj} \hat{g}_j^t$. Given that these are already identified as heterogeneous groups with conflicting gradient directions, why force integration of the projected gradients from these conflicting groups? While theoretically ensuring $\tilde{d}_k^t$ remains a descent direction, a clearer explanation is needed regarding the specific benefits this cross-group forced collaboration offers compared to completely isolating conflicting groups.

2．The paper proposes classifying clients into distinct groups based on their homogeneity levels. This approach of aligning optimization objectives is intuitively similar to existing Clustering Federated Learning (CFL) methods. The authors requires clearer definition of whether FedGCM constitutes a special case of CFL or maintains a globally shared knowledge space after grouping.

3．The core motivation hinges on resolving optimization conflicts driven by homophily heterogeneity using a single proxy scalar, $S_i$. However, in practical Non-IID federated networks, two clients might exhibit the exact same average homophily level but possess fundamentally different underlying feature distributions or semantic label spaces.

4．The effectiveness of RPGrad is highly sensitive to the collaboration strength hyperparameter $\eta$. Given that central servers in real-world FL deployments typically lack access to global validation sets, how to adaptively tune this critical parameter without manual intervention remains an unresolved practical challenge.

5．It remains unclear how much improvement on heterophilous datasets stems directly from FedGCM's gradient surgery versus the inherently strong heterophily-aware baseline.

6．While the results on 10 clients are promising, the practicality of the model in large-scale federated networks remains unproven.

---

> ### Author Rebuttal · Authors · 2026-03-29
>
> Thank you for your insightful comments.
>
> >Q1. Why force collaboration with conflicting groups?
>
> There may be misunderstanding: **RPGrad does not forcefully integrate conflicting gradients**. When two groups conflict, RPGrad first removes the conflicting component via orthogonal projection and retains the non-conflicting part.
> From a theoretical perspective, **complete isolation would block information sharing across subgraphs and may amplify local bias, which is contrary to the collaborative nature of FL**. From an empirical perspective, Figure 4 shows that **the w/ grouping variant consistently performs worse than FedGCM**, indicating that completely blocking cross-group information is suboptimal.
>
> >Q2. Difference with CFL
>
> We clarify that **FedGCM maintains a globally shared knowledge space after grouping, rather than learning isolated cluster-specific models as in standard CFL**. The main differences are as follows: 1) Standard CFL methods usually partition clients into disjoint clusters and train separate cluster-specific models with little or no cross-cluster interaction to mitigate Non-IID interference [1], whereas FedGCM explicitly preserves cross-group interaction through RPGrad in Eq. (9); 2) In CFL, clustering is the final objective for personalization, while in FedGCM, grouping is introduced to better avoid optimization conflicts under homophily heterogeneity.
>
> [1] Ghosh, et al., and Ramchandran, K. An efficient framework for clustered federated learning. NeurIPS, 2020.
>
> >Q3. …same homophily level but different….
>
> We agree that two clients may have the same $S_i$ while still differing in feature distributions or semantic label spaces, which can be viewed as a special case within our framework. Although grouping is based on $S_i$, **the actual local training is still driven by the cross-entropy loss on labeled nodes**. Therefore, feature and label differences can still affect the local gradients contributed by clients within a group. **Empirically, the consistent improvements of FedGCM across diverse datasets** suggest that our design remains effective, even when similar $S_i$ values may not imply identical semantic distributions.
>
> >Q4. Adaptive tuning of $\eta$
>
> As illustrated in Figure 3(a), FedGCM performs consistently well on most datasets when $\eta\in [0.2, 0.4]$. Specifically, a larger $\eta$ tends to work better on homophilous datasets, while a smaller $\eta$ is generally more suitable for heterophilous datasets. This is because $\eta$ controls the strength of cross-group information integration. On homophilous datasets, the structural discrepancy across groups may be relatively smaller, making stronger cross-group collaboration more likely to provide complementary benefits.
> **We agree that adaptive learning of $\eta$ remains unresolved, and we will discuss it in the limitations as an important direction for future work, as stated in our response to #Reviewer FEts-Q6.**
>
> >Q5. ...how much improvement...
>
> The gains on heterophilous datasets are **not solely** inherited from the heterophily-aware backbone. **As shown in Table 1**, both $FedAvg_{ACM}$ and FedGCM use the same ACM-GNN backbone, so their performance difference reflects the contribution of FedGCM beyond the backbone. Although $FedAvg_{ACM}$ already improves over $FedAvg_{GCN}$, FedGCM still brings further gains. This is further supported by the ablation results in Figures 4–5, where homophily-driven grouping and RPGrad consistently improve performance over the backbone.
>
> >Q6. Comparison on large-scale clients
>
> We further scale the number of clients to 50 and 100 on relatively large datasets, including **PubMed (P), Physics (Ph), Minesweeper (M), and Penn94 (Pe)**. For Penn94, we use Metis instead of Louvain in the 50- and 100-client settings, since Louvain produces empty clients at larger scales.
> The results in the tables below show that FedGCM still achieves the best performance in most cases, demonstrating its effectiveness and practicality beyond the 10-client setting.
>
>
> | Method   | P@50 | P@100 | Ph@50 | Ph@100 | Pe@50 | Pe@100 | M@50 | M@100 |
> |----------|-----:|------:|------:|-------:|-----:|------:|-----:|------:|
> | FedAvg   | 87.8 | 87.9  | 95.8  | 95.6   | 69.3 | 68.0  | 80.2 | 80.8  |
> | FedNova  | 85.2 | 84.6  | 94.1  | 93.9   | 67.2 | 68.2  | 79.4 | 81.4  |
> | FedPub   | 87.2 | 86.9  | 95.6  | 94.8   | 71.5 | 69.2  | 80.3 | 80.6  |
> | FedGTA   | 85.8 | 85.0  | 94.2  | 94.5   | 70.0 | 67.4  | 80.3 | 81.0  |
> | AdaFGL   | 87.8 | 88.4  | 96.0  | 95.6   | 74.1 | 74.0  | 81.6 | 82.1  |
> | FedTAD   | 88.1 | **89.4** | **96.2** | 95.1 | 70.2 | 69.2  | 80.3 | 80.5  |
> | FedHERO  | 82.8 | 84.5  | 95.4  | 95.2   | OoM  | OoM   | 82.6 | 80.9  |
> | FedSPA   | 86.7 | 87.1  | 95.1  | 95.3   | **74.7** | **74.2** | 82.4 | 81.3  |
> | Ours     | **88.6** | 88.8 | **96.2** | **95.9** | 74.2 | 74.0  | **84.3** | **82.6** |
>
>
> >Q7. Limitations
>
> Due to the space limitation, please refer to our response to **#Reviewer FEts-Q6** .

---

> > ### Author Rebuttal · Reviewer_HUtS · 2026-04-02
> >
> > Thanks for the response. My main concerns have been well addressed.

---

> > > ### Author Response · Authors · 2026-04-03
> > >
> > > Dear Reviewer HUtS,
> > >
> > > Thank you very much for your timely response and for your recognition of our rebuttal. Your valuable insights have greatly contributed to enhancing the quality of our work. We will carefully incorporate your suggestions into the revised manuscript to further improve the clarity and soundness of the paper.
> > >
> > > Thank you once again for your time and valuable feedback.
> > >
> > > Sincerely,
> > >
> > > Authors

---

### Official Review · Reviewer_5cgz · 2026-03-13

**Soundness:** 2
**Presentation:** 3
**Significance:** 3
**Originality:** 2
**Overall Recommendation:** 4
**Confidence:** 2

**Summary:**

This paper studies federated graph learning (FGL) when clients have graphs with different levels of homophily. Such heterogeneity causes local models to follow different optimization directions, leading to gradient conflicts during global aggregation and degrading performance. To address this issue, the authors propose FedGCM, a framework that mitigates optimization conflicts without increasing client-side model complexity. The method first groups clients based on their estimated homophily levels using spectral properties of graph signals. It then applies a residual projection–based gradient surgery (RPGrad) on the server to remove conflicting gradient components while incorporating useful information from other groups. Finally, the server performs group-oriented model updates and broadcasting. Experiments on several homophilous and heterophilous graph datasets show that FedGCM achieves better node classification performance than existing federated graph learning methods.

**Compliance With Llm Reviewing Policy:**

Affirmed.

**Final Justification:**

The rebuttal, particularly the additional analyses on gradient conflicts and efficiency, has helped strengthen the paper. While I still have some concerns regarding the realism of the federated setting, the experimental protocol follows prior work. Overall, the rebuttal has addressed several of my key concerns, and I have increased my score accordingly.

**Key Questions For Authors:**

Q1. If the graph originally belongs to a single dataset, how realistic is the federated scenario?

Q2. Do gradients from different client groups actually point in different directions during training? Could you provide empirical evidence, such as cosine similarity between gradients across groups?

Q3. Could the authors provide empirical comparisons of runtime and memory usage between the proposed method and existing baselines?

**Limitations:**

yes

**Strengths And Weaknesses:**

**Strengths**

S1. The paper is well written and generally easy to follow.

S2. The proposed framework is conceptually simple and easy to integrate into existing federated learning pipelines.

**Weaknesses**

W1. The paper motivates the method by arguing that clients with different homophily levels produce conflicting optimization directions, which leads to gradient conflicts during aggregation. However, the paper does not provide empirical evidence showing that gradients from different groups indeed point in significantly different directions. For example, it would be helpful to analyze the cosine similarity between gradients across client groups or visualize their distributions during training. Such an analysis could better support the core motivation of the proposed gradient surgery mechanism.

W2. While the paper provides a theoretical convergence analysis, it does not present a clear time and memory complexity analysis of the proposed method. In particular, the additional server-side operations, such as client grouping and gradient surgery across groups, may introduce non-negligible computational and memory overhead as the number of clients or model dimensions increases. A more explicit analysis of the scalability of the method in terms of runtime and memory usage would strengthen the paper.

---

> ### Author Rebuttal · Authors · 2026-03-29
>
> Thank you for your constructive comments.
>
> >Q1. Realism of the federated setting
>
> The realism of a federated scenario is determined by data ownership and privacy boundaries. In FGL, this may correspond to either multiple graph samples owned by each client or a large global graph distributed across clients as local subgraphs. For the node-level task considered in this work, **partitioning a complete graph via Louvain [1] or Metis [2] into multiple local subgraphs has become a widely adopted and accepted experimental protocol in prior FGL studies [3][4]. As described in the Implementation Details of Section 4.1**, we specifically adopted the Louvain method to simulate client environments with data heterogeneity. Compared to Metis, which prioritizes balanced partitioning for load balancing, Louvain better preserves the underlying community structure of the original graph.
>
> [1] Blondel, et al., Fast unfolding of communities in large networks. Journal of Statistical Mechanics: Theory and Experiment, 2008.
> [2] G. Karypis et al., “A fast and high-quality multilevel scheme for partitioning irregular graphs,” SIAM Journal on scientific Computing, 1998.
> [3] Fu, L., et al., Less is more: Federated graph learning with alleviating topology heterogeneity from a causal perspective. ICML, 2025.
> [4] Tan, Z.et al., Fedspa: Generalizable federated graph learning under homophily heterogeneity. CVPR, 2025.
>
> >Q2. Do gradients from different groups point in different directions during training? Cosine similarity between gradients across groups
>
> Yes, the gradients from different groups can point in different directions during training. To verify this, we compute the cosine similarity between gradients across groups on two homophilous (Cora and CiteSeer) and two heterophilous datasets (Chameleon and MineSweeper).
> **As shown in the following tables, we set $K=4$, and several group pairs exhibit negative cosine similarity**, directly indicating the presence of inter-group gradient conflicts. Overall, these statistics show that the gradient directions across groups are diverse, providing empirical evidence for the significance of considering inter-group optimization conflicts.
> | Datasets   | G0-1 | G0-2 | G0-3 | G1-2 | G1-3 | G2-3 |
> |-----------|------|------|------|------|------|------|
> | Chameleon | -0.045 | 0.043 | 0.226 | -0.065 | -0.120 | 0.156 |
> | Minesweeper | 0.274 | -0.144 | 0.035 | 0.157 | -0.144 | -0.205 |
> | Cora      | -0.013 | -0.111 | -0.167 | 0.067 | 0.066 | 0.124 |
> | Citeseer  | 0.150 | 0.086 | 0.059 | 0.006 | -0.005 | 0.126 |
>
> >Q3. Could the authors provide empirical comparisons of runtime and memory usage between the proposed method and existing baselines?
>
> Compared with existing multi-channel methods, FedGCM preserves client efficiency by shifting conflict mitigation to the server. Each client only maintains a standard backbone model without parameter duplication, thus **avoiding the substantial client-side memory overhead introduced by multi-channel architectures**. The additional cost of FedGCM mainly comes from gradient grouping and RPGrad on the server. Since conflict mitigation is conducted at the group level rather than exhaustively across all client pairs (see Section 4.4), the additional server-side overhead remains manageable.
>
> **We compare runtime (Time), client-side memory (Mem-C), and server-side memory (Mem-S) on PubMed and Amazon-ratings with 10 and 50 clients** under the same setting: 100 communication rounds, 5 local epochs, and a 3-layer ACM-GNN with hidden size 64. Client-side memory is reported as the average total memory across selected clients, while server-side memory is measured during aggregation, with total memory defined as peak GPU reserved memory plus peak CPU RSS increment.
> As shown below, FedAvg is the fastest and most memory-efficient due to its single-model client design and simple aggregation in the server side. **FedGCM achieves better performance than FedAvg (Table 1) while keeping client-side memory and computation close to the backbone and outperforming multi-channel baselines.**
> | PubMed | Time@10 | Mem-S@10 | Mem-C@10 | Time@50 | Mem-S@50 | Mem-C@50 |
> |--------|--------:|---------:|---------:|--------:|---------:|---------:|
> | FedAvg(backbone) | 38.1 | 232.0 | 304.2 | 173.7 | 148.1 | 161.5 |
> | FedGSP | 362.6 | 1413.8 | 838.0 | 3182.4 | 6310.4 | 957.0 |
> | AdaFGL | 86.1 | 276.2 | 330.5 | 400.0 | 226.0 | 174.8 |
> | FedSPA | 68.7 | 411.6 | 333.7 | 276.3 | 450.3 | 289.9 |
> | FedGCM | 41.2 | 252.7 | 308.1 | 258.9 | 169.3 | 164.4 |
>
> | Amazon | Time@10 | Mem-S@10 | Mem-C@10 | Time@50 | Mem-S@50 | Mem-C@50 |
> |--------|--------:|---------:|---------:|--------:|---------:|---------:|
> | FedAvg(backbone) | 42.0 | 294.0 | 380.0 | 195.2 | 98.2 | 114.0 |
> | FedGSP | 303.9 | 676.4 | 557.4 | 2164.2 | 3375.2 | 1116.5 |
> | AdaFGL | 75.9 | 340.3 | 398.8 | 530.2 | 174.5 | 153.5 |
> | FedSPA | 137.8 | 2009.1 | 1783.7 | 275.9 | 378.8 | 241.6 |
> | FedGCM | 42.7 | 314.3 | 383.3 | 271.9 | 118.8 | 116.0 |

---

> > ### Author Rebuttal · Reviewer_5cgz · 2026-04-04
> >
> > It remains unclear whether the datasets used for evaluation truly reflect data ownership and privacy boundaries in real-world federated settings. However, I acknowledge that the experimental protocol follows prior work. While some concerns still remain in this regard, other issues have been sufficiently addressed. I will increase my score.

---

> > > ### Author Response · Authors · 2026-04-07
> > >
> > > Dear Reviewer 5cgz,
> > >
> > > Thank you for your timely response and for your recognition of our rebuttal. We sincerely appreciate your decision to raise the score.
> > >
> > > While partitioning a complete graph into local subgraphs is a standard protocol in current FGL research, we acknowledge that this setting may not fully capture the complexity of real-world data ownership and privacy boundaries. We will clarify this scope in the final manuscript.
> > >
> > > Your suggestions are very valuable for improving the paper. Thank you once again for your time and constructive feedback.
> > >
> > > Sincerely,
> > >
> > > Authors

---

### Decision · Program_Chairs · 2026-04-30

**Decision:**

Accept (regular)

**Comment:**

This paper proposes a Federated Graph Learning framework by aligning inconsistent optimization objectives via a tailored gradient surgery scheme to address the challenge of homophily heterogeneity across clients. It utilizes the homophily level to divide nodes and employs residual projection to resolve inter-group interference. Theoretical and experimental investigations justify the superiority. All four reviewers provide positive feedback by recognizing the novelty and theoretical contributions. The concerns raised by the reviewers are fully resolved during the rebuttal.